# Revealing a hidden conducting state by manipulating the intracellular domains in $K_V10.1$ exposes the coupling between two gating mechanisms

Reham Abdelaziz[1], Adam P Tomczak[1], Andreas Neef[2]*, Luis A Pardo[1]*

[1]Oncophysiology Group. Max Planck Institute for Multidisciplinary Sciences, City Campus, Göttingen, Germany; [2]Neurophysics Laboratory, Göttingen Campus Institute for Dynamics of Biological Networks, Göttingen, Germany

*For correspondence:
aneef@gwdg.de (AN);
pardo@mpinat.mpg.de (LAP)

Competing interest: The authors declare that no competing interests exist.

**Abstract** The *KCNH* family of potassium channels serves relevant physiological functions in both excitable and non-excitable cells, reflected in the massive consequences of mutations or pharmacological manipulation of their function. This group of channels shares structural homology with other voltage-gated $K^+$ channels, but the mechanisms of gating in this family show significant differences with respect to the canonical electromechanical coupling in these molecules. In particular, the large intracellular domains of *KCNH* channels play a crucial role in gating that is still only partly understood. Using *KCNH1*($K_V10.1$) as a model, we have characterized the behavior of a series of modified channels that could not be explained by the current models. With electrophysiological and biochemical methods combined with mathematical modeling, we show that the uncovering of an open state can explain the behavior of the mutants. This open state, which is not detectable in wild-type channels, appears to lack the rapid flicker block of the conventional open state. Because it is accessed from deep closed states, it elucidates intermediate gating events well ahead of channel opening in the wild type. This allowed us to study gating steps prior to opening, which, for example, explain the mechanism of gating inhibition by $Ca^{2+}$-Calmodulin and generate a model that describes the characteristic features of *KCNH* channels gating.

## eLife assessment

This **valuable** study examines the role of the interaction between cytoplasmic N- and C-terminal domains in voltage-dependent gating of Kv10.1 channels. The authors suggest that they have identified a hidden open state in Kv10.1 mutant channels, thus providing a window for observing early conformational transitions associated with channel gating. The evidence supporting the major conclusions is **solid**, but additional work is required to determine the molecular mechanism underlying the observations in this study. Learning the molecular mechanisms could be significant in understanding the gating mechanisms of the KCNH family and will appeal to biophysicists interested in ion channels and physiologists interested in cancer biology.

## Introduction

Voltage-gated potassium channels constitute a large family of proteins that allow $K^+$ flow upon changes in the membrane potential. Their general architecture consists of a tetrameric complex with six transmembrane segments (S1-S6) in each subunit to form a central pore with four voltage sensors at the periphery. S1 to S4 segments constitute the sensor domain, while S5 and S6 segments, together with

the loop linking them, line the pore. The mechanism of voltage-dependent gating is well understood for several subfamilies, namely those closely related to the *Drosophila Shaker* channel (*KCNA* and *KCNB*). In this subset of families, the linker between the sensor and pore domains (S4-S5 linker) acts as a mechanical lever transferring the movement of the voltage sensor to the gate at the bottom of S6 in a neighboring subunit through physical interaction (*Barros et al., 2020*). Such trans-subunit interaction is commonly denoted 'domain swapping'. In other families, like the EAG family (*KCNH*; *Malak et al., 2019*; *Tomczak et al., 2017*; *Whicher and MacKinnon, 2019*), gating mechanics must be different because there is no domain swapping in the transmembrane regions, the S4-S5 segment is very short and does not form a helix (*Whicher and MacKinnon, 2016*; *Wang and MacKinnon, 2017*), and voltage-dependent gating occurs even when the S4-S5 linker is severed or removed in $K_V$10.1 (*Lörinczi et al., 2015*) or $K_V$11.1 (*de la Peña et al., 2018*). A thorough understanding of gating mechanisms in this family is also relevant in practical terms because members of the family are implicated in many pathological conditions, as is the case for Kv101 (*Bauer and Schwarz, 2018*; *Toplak et al., 2022*), which makes these channels attractive therapeutic targets.

Although *KCNH* channels do not display domain-swapping in the transmembrane domains, they show extensive conserved intracellular domains where intersubunit interactions occur. The so-called eag domain in the N-terminus, formed by the PAS domain and PAS-Cap, interacts with the CNBHD (cyclic nucleotide-binding homology domain, see *Figure 1—figure supplement 1A*) of the neighboring subunit, which is connected to the S6 through the C-linker. The four eag-CNBHD complexes form a ring in the intracellular side of the channel, connected to the gate via the C-linkers (*Whicher and MacKinnon, 2019*; *Whicher and MacKinnon, 2016*). Thus, there is domain-swapping within intracellular domains instead of the transmembrane segments (reviewed, e.g. in *Barros et al., 2020*; *Codding et al., 2020*). This arrangement makes the intracellular ring an excellent candidate to participate in the gating process, as it has been demonstrated for other channels (e.g. *Codding et al., 2020*; *James and Zagotta, 2018*; *Núñez et al., 2020*; *Verkest et al., 2022*). The first residues of the PASCap segment, although unresolved in the available structures of Kv10.1, occupy in Kv11.1 a space close to the S4-S5 linker and the C-linkers, suggesting its participation in transducing the movements of the voltage sensor to the intracellular ring (*Wang and MacKinnon, 2017*; *Stevens-Sostre et al., 2024*). Indeed, the gating of *KCNH* channels is affected by manipulations of either the eag domain, the CNBHD, or the interaction between them (*Stevens-Sostre et al., 2024*; *Malak et al., 2019*; *Whicher and MacKinnon, 2019*; *Codding et al., 2020*; *Codding and Trudeau, 2019*; *Dai and Zagotta, 2017*; *Dai et al., 2018*; *Gianulis et al., 2013*; *Ju and Wray, 2006*; *Sahoo et al., 2012*; *Terlau et al., 1997*). Furthermore, the interaction between the PAS domain and the C-terminus is more stable in closed than in open $K_V$11.1 (HERG) channels, and a single chain antibody binding to the interface between PAS domain and CNBHD can access its epitope in open but not in closed channels, strongly supporting a change in conformation of the ring during gating (*Harley et al., 2021*). Recent cryo-EM work revealed that the position of the voltage sensor in a polarized membrane would preclude the opening of the gate (*Mandala and MacKinnon, 2022*). After relieving such an obstacle upon depolarization, a rotation of the intracellular ring would allow the helices at the bottom of S6 to dilate, opening the gate (*Whicher and MacKinnon, 2019*; *Whicher and MacKinnon, 2016*; *Wang and MacKinnon, 2017*; *Wynia-Smith et al., 2008*). Moreover, cryo-EM data on $K_V$10.2 (which shows over 75% identity with $K_V$10.1) reveals a pre-open state in which the transmembrane regions of the channel are compatible with ion permeation (the permeation path is dilated, like in open states) but the intracellular gate is still in the same conformation as in closed states (*Zhang et al., 2023*). Whether the difference between these two states depends on the rotation of the intracellular ring remains to be addressed.

The intracellular ring is also subject to modulation by factors external to the channel molecule. For example, ligand binding to the PAS domain can modify gating (*Wang et al., 2020*; *Wang et al., 2023*), explaining the complex effects induced by some inhibitors such as imipramine. The association of $K_V$10.1 with calmodulin (CaM), which efficiently inhibits the channel in the presence of $Ca^{2+}$, adds an additional level of complexity (*Whicher and MacKinnon, 2016*; *Schönherr et al., 2000*). $K_V$10.1 has CaM binding sites at the N- and the C-termini (*Gonçalves and Stühmer, 2010*; *Schönherr et al., 1999*; *Ziechner et al., 2006*). CaM presents two $Ca^{2+}$-binding lobes (N- and C-lobe) joined by a flexible helix. The N-lobe of CaM (*Babu et al., 1985*) binds to the N-terminal site in one channel subunit, while the C-lobe binds the C-terminal sites of the opposite subunit (*Whicher and MacKinnon, 2016*),

therefore linking two PAS/CNBHD pairs. It is plausible that cross-linking of two segments of the ring impairs its rotation and can thus be the basis of the inhibitory action of CaM.

Yet, the mechanism coupling the movement of the voltage sensor and the rotation of the ring remains unsolved. A candidate to fulfill this function is the segment encompassing the first ten N-terminal residues, which, by analogy to Kv11.1, could interact with the distal end of S4 (*Stevens-Sostre et al., 2024*). Importantly, mutations in both the intracellular N- (*Whicher and MacKinnon, 2019*) and C-terminal domains (*Zhao et al., 2017*) cause unexpected rectification or paradoxical activation by Ca-CaM instead of the normal inhibition seen in wild type (WT) (*Lörinczi et al., 2016*).

In this study, we provide a mechanistic explanation of the hallmarks of $K_V10.1$ gating and define the role of the intracellular ring using a combination of electrophysiology and mutagenesis, modeling, and biochemical approaches. Channel variants with altered interaction between PASCap and CNBHD uncovered an open state in the mutants, different from the main open state in the WT. Unlike the canonical open state, the mutant-specific open state is not subjected to a flicker block and, therefore, gives rise to a larger macroscopic current. Because this state is best accessed from deep closed states (strong hyperpolarized membrane potential), it made the transition between the voltage sensor movement and the ring rotation visible, allowing the characterization of gating steps hidden in WT. Furthermore, access to the novel open state is promoted by the binding of CaM, thus explaining the paradoxical effects of CaM in mutant channels. We propose a model tightly constrained by data that can explain the main differential features of $K_V10.1$ gating.

## Results

### Disrupting the interaction between PASCap and CNBHD reveals a biphasic gating behavior

The first N-terminal residues of $K_V10.1$ (PASCap *Morais Cabral et al., 1998*) are prime candidates for transmitting voltage sensor motion to the intracellular domain based on crystal structures and interactions inferred from mutagenesis experiments. In particular, two residues at the bottom of S4 –H343 (*Terlau et al., 1997*) and D342 (*Tomczak et al., 2017*)– functionally interact with the initial N-terminus. The N-terminal PAS domain can then transmit mechanical cues to its C-terminal interaction partner CNBHD (*Whicher and MacKinnon, 2019*; *Whicher and MacKinnon, 2016*; *Codding et al., 2020*; *Codding and Trudeau, 2019*; *Mandala and MacKinnon, 2022*; *Wang et al., 2020*). To study $K_V10.1$ gating under perturbed intramolecular interaction, we deleted the PASCap domain (residues 2–25) and, in a more conservative approach, we examined the impact of a point mutation (E600R, *Figure 1—figure supplement 1B*) reported to disrupt the PASCap-CNBHD interaction (*Haitin et al., 2013*) and which shows a behavior reminiscent of the PASCap deletion in its response to calmodulin (see below; *Lörinczi et al., 2016*). We then obtained the response of the mutants to discrete depolarizations to different potentials (–100 to +120 mV for 300ms) in *Xenopus* oocytes in the presence of 60 mM $K^+$ in the extracellular solution to follow deactivation behavior through tail currents.

*Figure 1A* displays representative current traces of WT and the mutant channels. The threshold for activation was shifted towards hyperpolarizing values in both mutants, giving rise to inward currents at potentials that do not lead to the opening of WT channels. This shift is most evident in the biphasic conductance-voltage (GV) plots (*Figure 1B*; *Figure 1—source data 1*). The kinetics of activation (*Figure 1C*) and deactivation (*Figure 1E*) of the mutants were also clearly distinguishable from WT. To facilitate the description of the results, we classified the responses into three categories depending on the stimulus amplitude: *weak* (–90 mV to − 20 mV), *moderate* (–10 mV to +40 mV), and *strong* (+50 to+120 mV).

While the activation of WT currents does not accelerate dramatically with increasing depolarizations in the voltage range tested, the mutants activated much slower than the WT upon *weak* and *moderate* depolarizations, reaching activation kinetics similar to WT with *strong* ones, as can be observed in normalized traces (*Figure 1C*). To obtain a more quantitative estimation of the changes in activation velocity, we used the time required to reach 80% of the maximum current amplitude plotted against the stimulus voltage (*Figure 1D*; *Figure 1—source data 2*). ΔPASCap and E600R needed a much longer time than WT to activate at *weak* depolarizing potentials but were equally fast at membrane potentials larger than +50 mV.

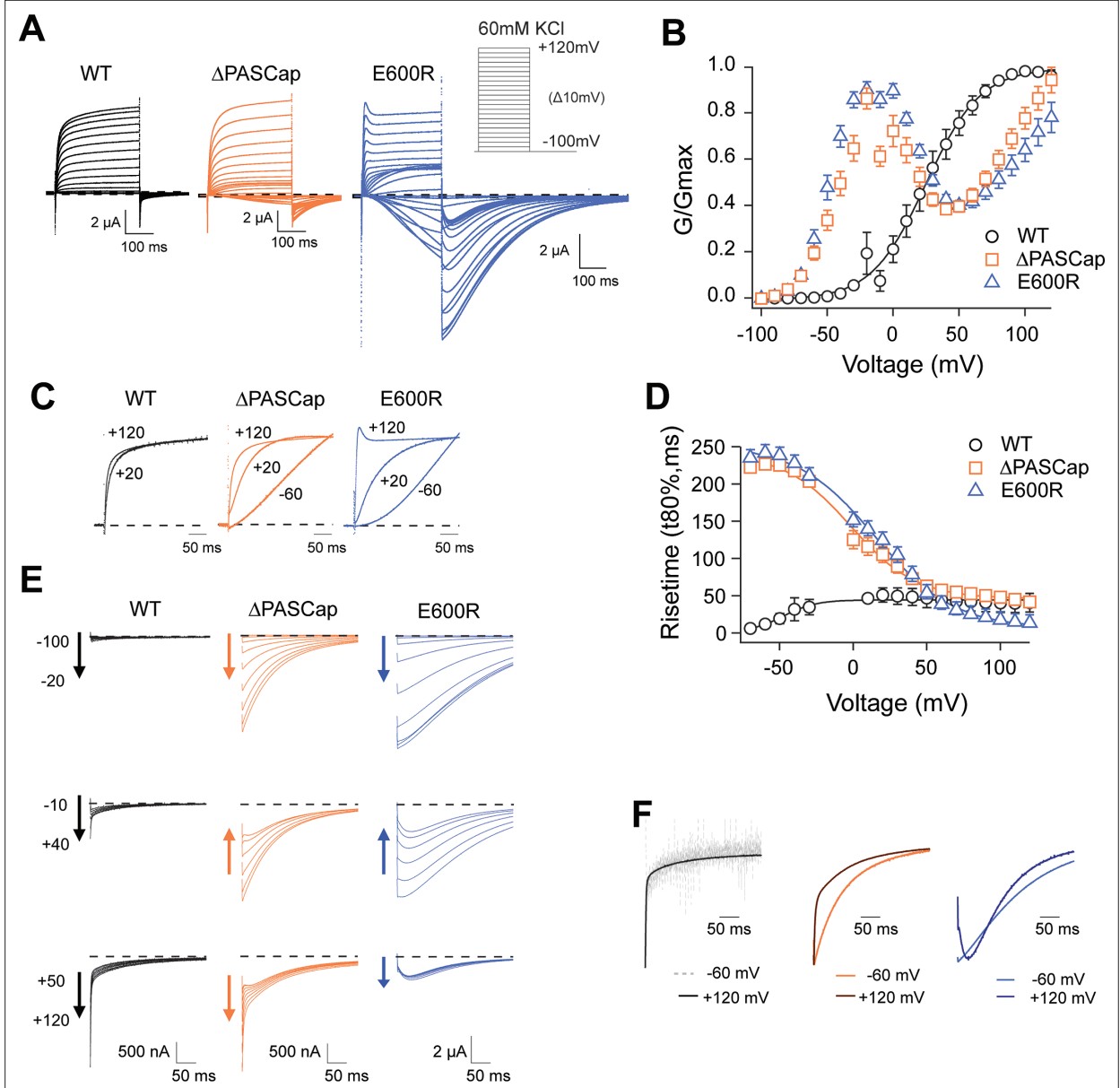

**Figure 1.** Characterization of ΔPASCap and E600R mutants. (**A**) Raw current traces resulting from depolarizations between –100 and +120 mV in WT (black), ΔPASCap (orange), and E600R (blue). The dashed lines indicate zero current. The stimulus protocol is schematically depicted in the inset. The extracellular solution contained 60 mM KCl, 57.5 mM NaCl, 10 mM HEPES, 1.8 mM CaCl2, pH = 7.4 (**B**) GV plots corresponding to the three channel variants (colors as in A). (N: WT = 6, ΔPASCap = 7, E600R=11; mean ± SEM) (**C**). Normalized traces to the indicated voltages to reveal the acceleration of activation with depolarization in the mutants. WT does not give rise to significant outward current at –60 mV. (**D**) Rise time of ΔPASCap and E600R as a function of voltage (colors as in A). The activation is much slower than in WT up to +50 mV but reaches the speed of WT with stronger stimuli. (**E**) Tail currents at –100 mV after depolarizations to potentials in the weak, medium, or strong range (up to down, see text for details). The arrows indicate the direction of the change in tail peak amplitude with increasing voltage. (**F**) Normalized tail currents at –100 mV after depolarizations to the indicated voltages.

The online version of this article includes the following source data and figure supplement(s) for figure 1:

**Source data 1.** Stimulus voltage and individual current amplitudes related to *Figure 1B*.

**Source data 2.** Stimulus voltage and individual rising times related to *Figure 1D*.

**Figure supplement 1.** Structure of the cytoplasmatic ring of Kv10.1.

The deactivation (tail) kinetics changes with increasing depolarizations were more evident and more complex than changes in activation kinetics (*Figure 1E and F*). For increasing test pulse potentials, the peak amplitude of ΔPASCap tail currents first increased progressively, then decreased at *moderate* values, and rose again in the range of *strong* depolarizations. E600R showed a similar pattern of tail amplitude, while the increase at *strong* potentials was less pronounced although still evident. Remarkably, both the tail amplitude and its decay kinetics underwent profound changes depending on the stimulus. While the kinetics of WT tail currents was the same across different potentials, showing the characteristically fast deactivation of $K_V10.1$, ΔPASCap deactivated slow- and monotonically at *weak* depolarizations, but a fast component started to become evident after *moderate* stimuli. The fast component dominated the process at *strong* depolarizations. For E600R, the deactivation after *weak* stimuli was also slow and accelerated after a rising phase in the *moderate* and *strong* depolarization range.

Due to the complex behavior of the tail currents, different equations would be needed to fit the tails of the various channels to extrapolate the amplitude to time zero. Hence, to calculate the macroscopic conductance, we simply used the current amplitude at the end of the depolarizing stimulus and divided it by the driving force calculated from the measured reversal potential (*Figure 1B*). As already observed in the raw traces, the threshold for activation for both mutants was strongly shifted in the hyperpolarizing direction with respect to WT (–80 mV *vs.* –40 mV). Still, the most evident change was that both ΔPASCap and E600R displayed a biphasic GV in contrast to WT. *Weak* depolarizing pulses increased the macroscopic conductance of both mutants until a maximum at approximately +10 mV. With further depolarizations, the conductance initially declined to rise again in response to *strong* depolarizations. This finding matches the changes in amplitude of the tail currents, which, therefore, probably reflect a true change in overall conductance. A similar behavior has been mentioned for related (*Whicher and MacKinnon, 2019*) or unrelated mutations (*Zhao et al., 2017*) affecting the intracellular domains. However, the reasons for this phenomenon have not been investigated. We thus aimed to understand the molecular mechanisms underlying the biphasic GV.

## The biphasic GV is described by two sigmoidal components corresponding to a two-step gating mechanism

One possible explanation for the biphasic behavior could be the coexistence of two separate channel populations with different kinetics, amplitude, and voltage dependence. This seems unlikely because the shape of the GVs was consistent in all our recordings, as evidenced by the small error bars (*Figure 1B*) despite the variability intrinsic to the oocyte system. Alternatively, each channel could have two open states, and the rectification observed between the two components could represent a transition from one state to the other. Indeed, an equation that reflects the two components and a transition between them (*Equation 3b*; see Materials and methods) described the behavior of the GV of both mutants accurately (*Figure 2A*; *Figure 2—source data 1*). Representative current traces are shown in (*Figure 1A* and *Figure 2A and C*).

With the available structural information in mind, the two components could represent sequential access to two open states (from here on, $O_1$ and $O_2$) through separate gating steps that differentially involve the sensor movement and ring rotation (*Tomczak et al., 2017*; *Whicher and MacKinnon, 2019*; *Mandala and MacKinnon, 2022*).

To test if the ring underlies one of the two gating steps, we tested the behavior of additional N-terminal deletions of increasing length (Δ2–10, 2–25 for ΔPASCap, and 2–135 for Δeag), expected to disrupt the ring integrity more and more. A biphasic GV was observed in all these mutants (*Figure 2A*). The $V_{half}$ value of the first component was very similar across mutants. In contrast, the second component showed different thresholds. We then performed a global fit using *Equation 3b*, where we linked the parameters of the first components ($Vh_1$, $K_1$) across mutants and allowed the parameters for the second component ($Vh_2$, $K_2$, $A_2$) and the transition ($Vh_3$, $K_3$) to vary (*Supplementary file 1*, *Figure 2B*; *Figure 2—source data 2*). The global fit accurately described the behavior of the GV in all mutants (*Figure 2A*).

The global fit results indicate that the first component is conserved across mutants. In contrast, the second component occurs at progressively more depolarized potentials for increasingly larger N-terminal deletions and when the structure of the ring is altered through disruption of the interaction between N- and C-termini (E600R). Thus, the underlying gating events can be separated into two

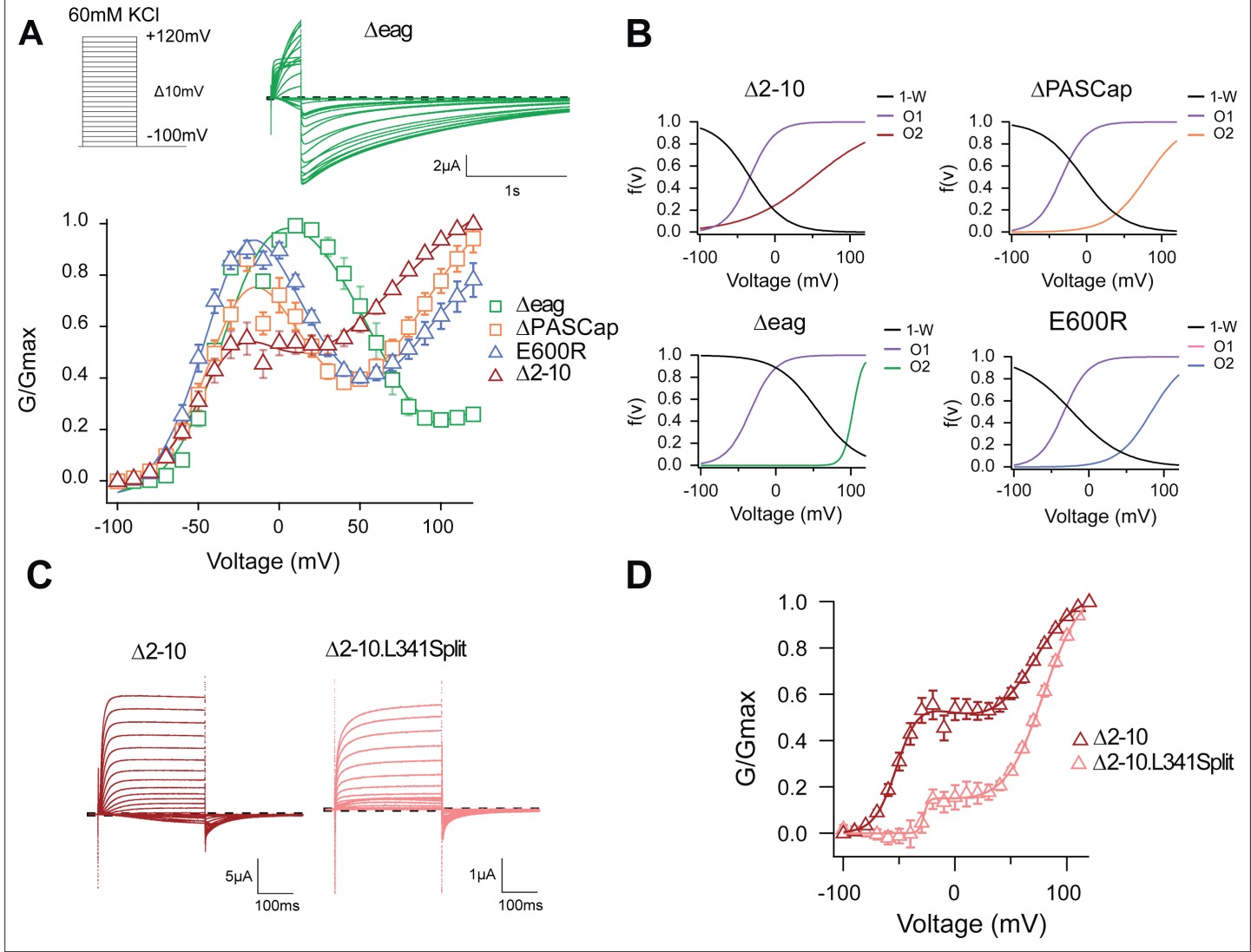

**Figure 2.** The biphasic GV corresponds to two sequential events. (**A**) The GV of all tested mutants show biphasic behavior. (N: Δ2–10=6, ΔPASCap = 7, Δeag = 7, E600R=11; mean ± SEM). All are well described by *Equation 3b* in a global fit with fixed parameters for the first component. (**B**) Distribution of the three components used for the fits as a function of voltage. (**C**). A discontinuous form of Δ2–10 shows attenuated biphasic behavior. (**D**) GV plots of Δ2–10 and Δ2–10.L341Split (N: Δ2–10=6, Δ2–10.L341Split = 5; ± SEM). Split fitted using *Equation 3b*. The dashed lines in **A** and **C** indicate zero current. The stimulus protocol is schematically depicted in the inset in **A**. The extracellular solution contained 60 mM KCl.

The online version of this article includes the following source data for figure 2:

**Source data 1.** Stimulus voltage and individual current amplitudes related to *Figure 2A*.

**Source data 2.** Stimulus voltage and individual current amplitudes related to *Figure 2D*.

steps: The first gating step involves only the voltage sensor without engaging the ring and leads to a pre-open state, which is non-conducting in the WT but conducting in our mutants. The second gating event operates at higher depolarizations, involves a change in the ring, and leads to an open state both in WT and in the mutants.

To test this hypothesis, we compared the behavior of Δ2–10 and Δ2–10.L341Split, a channel lacking a covalent connection between the sensor and the pore domain (*Tomczak et al., 2017*), hence decoupling residues downstream S4 (in this case starting from 342) from the movement of the sensor. This decoupling would minimize the action of the displacement of the voltage sensor on the pore domain but would not necessarily affect the subsequent ring rotation. As predicted, compared to Δ2–10, Δ2–10.L341Split showed a significant reduction in the first component of the biphasic GV (*Figure 2C and D*). This indicates that the coupling of the voltage sensor motion to the entry into $O_2$ does not

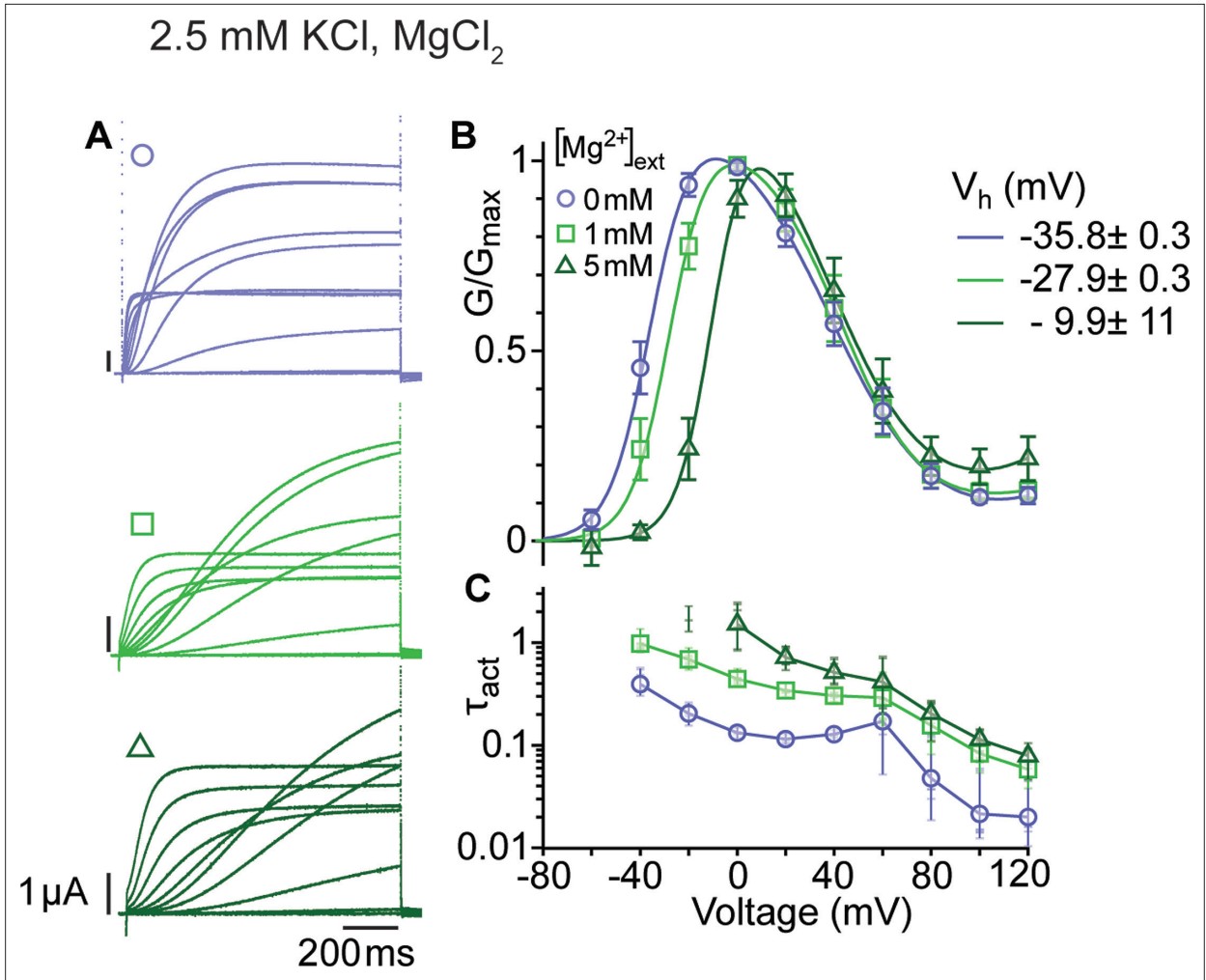

**Figure 3.** $Mg^{2+}$ induces a shift of the first component in the depolarizing direction. (**A**) Raw current traces from oocytes expressing Δeag channels in response to depolarizations from a holding potential of –100 mV to voltages between –100 and +120 mV in the presence of 0, 1 or 5 mM $MgCl_2$ in the external solution (legend in B). Scale bars are 1µA and 200ms. (**B**) Average GV plots (with SEM) obtained under the same conditions from n=6, 6, and 5 recordings (0, 1, and 5 mM $MgCl_2$). The $V_{half}$ of the first component shifts from –35.8 mV in the absence of extracellular $Mg^{2+}$ to –27.9 and –9.9 mV in 1 mM and 5 mM $MgCl_2$. (**C**). Time constants of the early activation show the decelerating effect of $Mg^{2+}$. Symbols depict averages, vertical lines the range and horizontal lines individual experiments. For voltages where conductances or time constants could not be reasonably estimated no data are displayed. The recordings were performed in an external solution containing 115 mM NaCl, 1.8 mM $CaCl_2$, 2.5 mM KCl, 10 mM Hepes pH 7.2, and the indicated concentrations of $MgCl_2$.

The online version of this article includes the following source data for figure 3:

**Source data 1.** Stimulus voltage and individual current amplitudes related to *Figure 3A*.

**Source data 2.** Stimulus potential and individual values of activation time constant related to *Figure 3B*.

require a continuous S4-S5 linker. If, in non-split mutants, the upward transition of S4 allows entry to $O_1$, it is reasonable to assume that the movement is not transmitted the same way in the split and the transition into $O_1$ is less probable. The observation that, in the split, entry into $O_1$ requires higher depolarization and appears to be less likely, suggests that downstream of S4 (beyond position 342), there is a mechanism to convey S4 motion to the gate of the mutants.

Activation of WT $K_V10.1$ channels (best studied for the *Drosophila* form *eag*) drastically slows down in the presence of extracellular divalent cations that bind to the voltage sensor (*Silverman et al., 2000*; *Tang et al., 2000*), for example $Mg^{2+}$, $Ni^{2+}$, $Co^{2+}$, and $Mn^{2+}$, but not $Ca^{2+}$. Strikingly, the degree of deceleration correlates with the ions' hydration enthalpy, suggesting that the ion might unbind during activation (*Terlau et al., 1996*). We chose to study the impact of $Mg^{2+}$ on the kinetics and voltage dependence of activation (*Figure 3*; *Figure 3—source data 1*&*2*) in the mutant Δeag, which

shows the most significant separation between the two conductance components. $Mg^{2+}$ slowed the activation of the channel at all depolarizations, affecting entry to both open states. This result suggests that $Mg^{2+}$ also slows the voltage sensor motion that controls access into the second open state. In addition, the voltage dependence of activation was shifted by approximately 25 mV to more depolarized voltages. This is in line with deep deactivated states bound to $Mg^{2+}$ and requiring stronger depolarization to exit them. Interestingly, the transition from the first to the second component of the GV plot seems unaffected by the extracellular divalent binding, which could be expected if the transition occurs in the cytoplasm.

## Steady-state voltage dependence and activation kinetics are consistent with two open states ($O_1$ and $O_2$) with different conducting behavior

Thus far, our results are compatible with two different open states in the mutants. The first open state, the mutant-specific $O_1$, would dominate at *weak* depolarizations and deactivate slowly, with tail current decay time constants of tens of milliseconds. In contrast, mutant and WT channels would

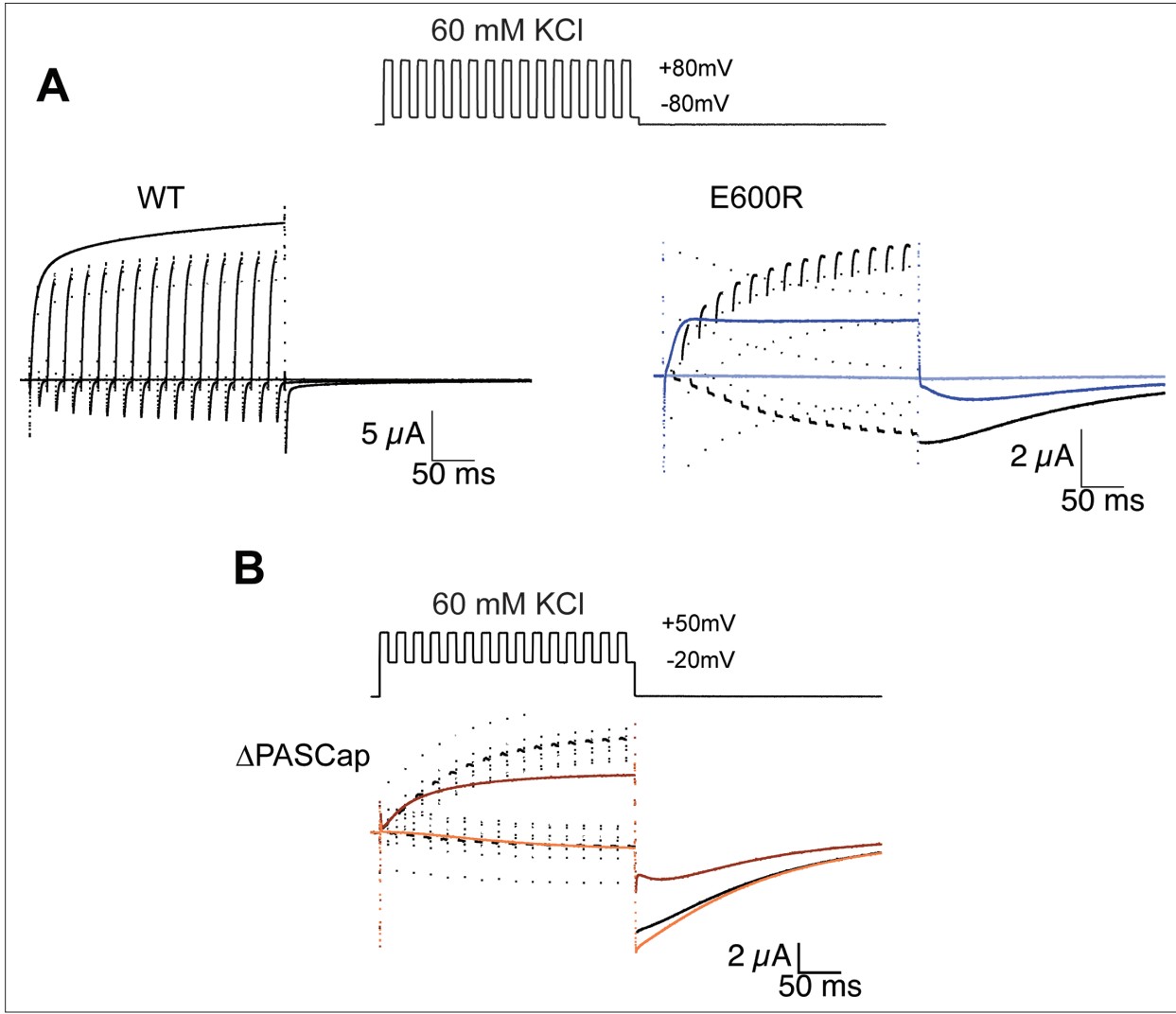

**Figure 4.** Alternating stimuli reveal larger macroscopic conductance for $O_1$. (**A**) Alternating potential between –80 and +80 mV in the WT results in current amplitudes that are smaller than those during a sustained stimulus (Upper left traces). In contrast, E600R gave rise to larger currents when the stimulus was intermittent and too short to allow occupancy of $O_2$ (upper right). Current traces in response to a sustained stimulus are shown in blue (dark trace +80 mV; light trace is –80 mV) (**B**) The effect was qualitatively similar for ΔPASCap, which consistently gave rise to larger current upon oscillating stimuli between –20 and +50 mV than during a constant pulse to +50 mV. Current traces in response to sustained stimulus are shown in orange (dark trace +50 mV; light trace is –20 mV). The stimulus protocol for A and B is schematically depicted in the inset. The extracellular solution contained 60 mM KCl.

reach the second open state ($O_2$) at strong depolarizations and deactivate more rapidly, with a few milliseconds or less time constants. A critical test of this hypothesis is the application of more complex voltage stimuli that drive the system to a non-equilibrium state of high $O_1$ occupancy. This might be achieved by deactivation periods of around 10ms, just long enough to remove most channels from $O_2$ but sufficiently brief to maintain the occupancy of $O_1$. For simplicity, we used depolarization periods of the same duration and tested whether a 300ms series of activating and deactivating 10ms pulses could accumulate channels in a high conducting state $O_1$.

For WT (*Figure 4A*), alternating between –80 and +80 mV resulted in a smaller amplitude than a constant stimulus to +80 mV, just as expected for a system with a single open state and a monotonic voltage dependence of activation. Moreover, WT's rapid deactivation resulted in near-complete deactivation during every cycle.

In E600R, in contrast, the current amplitude during activating pulses increased steadily from cycle to cycle. Ultimately, the current amplitude exceeds that obtained with a constant +80 mV pulse (*Figure 4A*). Because the deactivation of this mutant is much slower and occurs at more negative potentials than that of WT (see *Figures 1 and 2B*), there was no noticeable deactivation during the –80 mV episodes. In addition, the tail current after the alternating stimuli had a larger amplitude than the one after a continuous pulse and decayed mostly as one component. All those observations are consistent with an accumulation of channels in $O_1$. A similar behavior was detected with ΔPASCap (*Figure 4B*), albeit within a different voltage range, as could be predicted from the different voltage dependence of the transition between states (*Figure 2B*). Again, alternating pulses resulted in larger current amplitudes and less complex tail current as compared to a single long pulse (*Figure 4B*).

## Single channel recordings reveal two distinct voltage-dependent behaviors in ΔPASCap

A larger current amplitude in $O_1$ can be due to a larger single-channel conductance or a different open probability. We performed single-channel recordings in membrane patches excised from oocytes expressing the ΔPASCap mutant to distinguish between these two possibilities. To limit the possible contamination by other channels, we measured the activity in outside-out patches with $K^+$ as the only cationic charge carrier on the cytoplasmic side (see Materials and methods), rendering a theoretical Nernst equilibrium potential for $K^+$ of –93.77 mV and –4.69 mV for $Cl^-$. We only considered channels producing outward currents at –60 mV, ruling out $Cl^-$ contamination. In such patches, we detected $K^+$ current during single-channel openings that typically lasted several milliseconds upon weak depolarizations but could rarely be resolved at strong depolarizations (*Figure 5A*). Such behavior was never detected in uninjected oocytes. The activity in the patches was silenced by the addition of 100 μM astemizole (*Figure 5—figure supplement 1*), providing robust evidence that the activity is the result of the expression of ΔPASCap.

To characterize the activity in a quantitative way, we performed series of sweeps where the patch was held at –100 mV for 10 s, depolarized to +10 mV for 2.5 s, held again at –100 mV for 5 s, and finally depolarized to +40 mV for 2.5 s. The procedure was repeated tens of times on each patch, and representative traces are shown in *Figure 5B*. At +10 mV, we detected openings lasting up to hundreds of milliseconds, while the dwell times were much shorter at +40 mV. In some cases, openings of the two classes were observed in the same sweep, with the long openings preferentially preceding the shorter ones. Any transition from short openings to long openings (or back) always involved a longer intermediate close time, indicating that the channel travels between the two states through one or more closed states without a direct link between the two open states.

## Deep-closed states favor access to $O_1$

A hallmark of $K_v10.1$ gating is the change in activation kinetics in dependence on the pre-pulse potential reminiscent of (and commonly named) the Cole-Moore Shift (*Cole and Moore, 1960*; *Hoshi and Armstrong, 2015*). Hyperpolarized potentials drive the channel into deep closed states, which delays and decelerates activation (*Ludwig et al., 1994*). This is well described by a model with four identical, independent transitions of the voltage sensor (*Schönherr et al., 1999*) and is compromised by N-terminal deletions (*Whicher and MacKinnon, 2019*). To test the behavior of our mutants concerning this property, we applied a series of 5s-long conditioning pulses with voltages ranging from –160 mV to –20 mV, followed by a test pulse to +40 mV in the absence of external $Cl^-$. The activation kinetics,

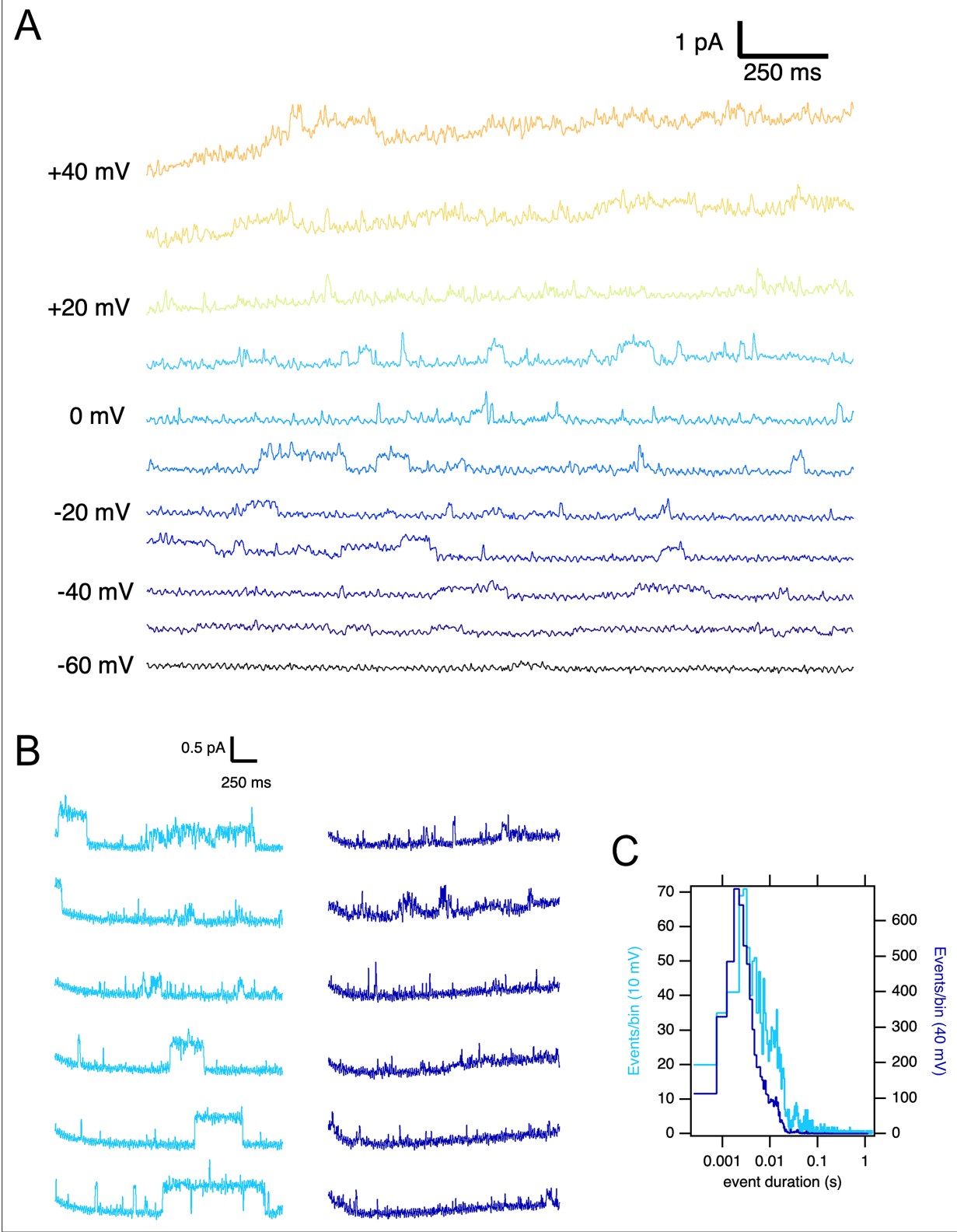

**Figure 5.** Single ΔPASCap channels reveal longer open times at moderate depolarizations. (**A**) Representative current traces at the indicated voltages obtained from a holding potential of –100 mV. (**B**) Comparison between traces obtained at +10 mV (light blue) and +40 mV (dark blue). The time spent in the open state is much longer at +10 mV. (**C**). Open time distribution for events shorter than 50ms obtained from currents in an experiment like the one depicted in B (100 stimuli at each potential).

*Figure 5 continued on next page*

*Figure 5 continued*

The online version of this article includes the following figure supplement(s) for figure 5:

**Figure supplement 1.** Single channel activity in an outside-out patch from an oocyte expressing ΔPasCap.

quantified by the time to 80% of the maximum current, showed a strong pre-pulse dependence in WT, ΔPASCap, and E600R, with much larger rise times in both mutants (*Figure 6B*; *Figure 6—source data 1*).

In the mutants, not only activation kinetics but also current amplitude was substantially affected by hyperpolarizing pre-pulses. With respect to the −100 mV pre-pulse potential, the current starting from a −160 mV pre-pulse increased in ΔPASCap by a factor of 3.93±0.36 and in E600R by a factor of 2.65±0.54. In contrast, WT current amplitude was not significantly affected (*Figure 6A and C*; *Figure 6—source data 2*). Such increases in amplitude are often related to augmented channel availability due to voltage-dependent recovery from inactivation. Still, although under some conditions, a

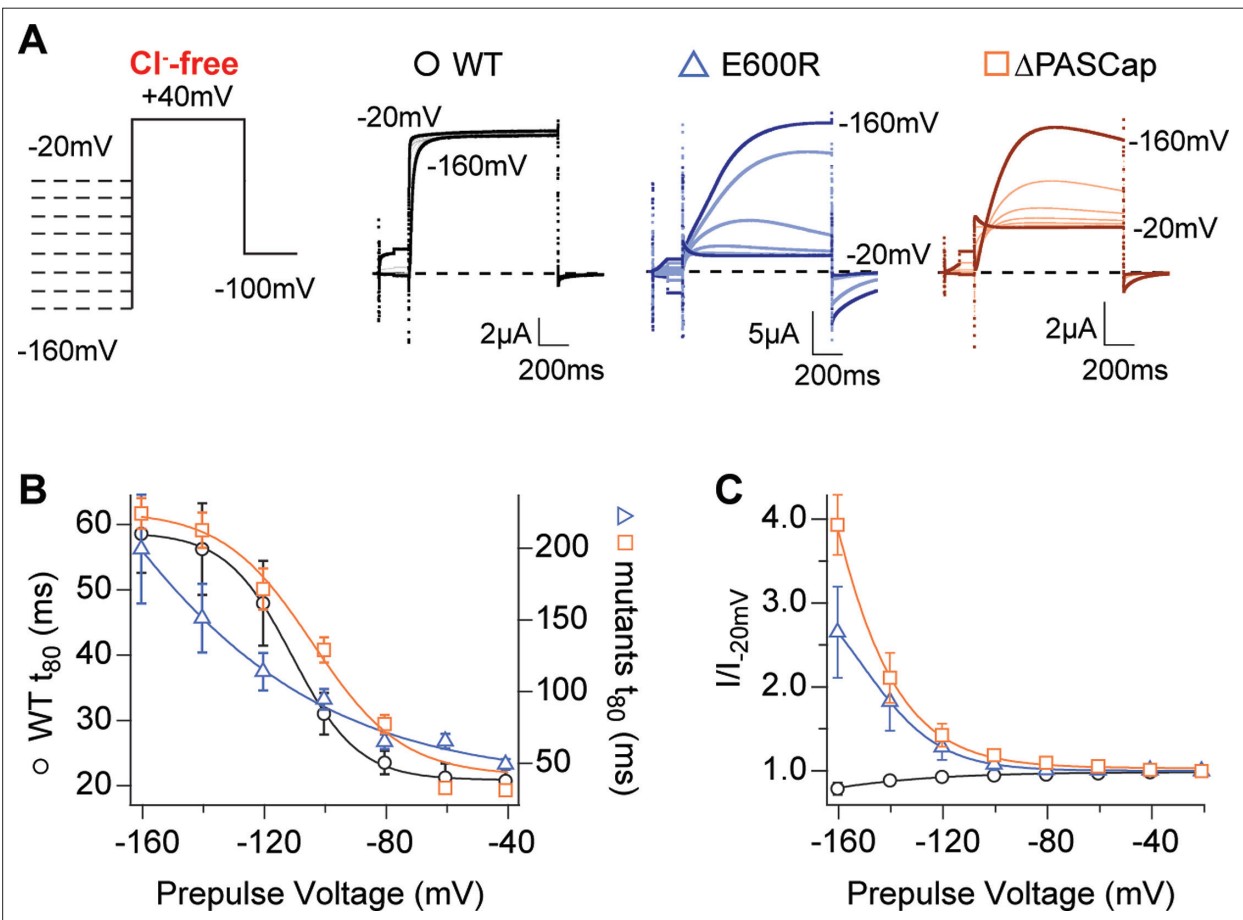

**Figure 6.** Hyperpolarization promotes access to a large conductance, slowly activating open state. (**A**) Raw current traces in response to the stimuli depicted in the scheme. (**B**) The rise time to 80% of the maximal current during the depolarizing stimulus is plotted *vs.* prepulse voltage (N: WT = 7, ΔPASCap = 9, E600*R*=8; mean ± SEM). Although the activation is much slower for both mutants (note the different y axis for the mutants), they retain a strong dependence on the prepulse potential. (**C**) Normalized end-pulse current (I/I$_{-20}$) is plotted *vs.* prepulse voltage (N: WT = 10, ΔPASCap = 8, E600*R*=8; mean ± SEM). The amplitude of the current at +40 mV increased markedly when the holding potential was below −100 mV in the mutants, while the amplitude in WT changed only marginally. The dashed lines indicate zero current. The stimulus protocol is schematically depicted in the inset. A 5 s pre-pulse was applied, of which only the initial and last 100ms were recorded. The chloride-free extracellular solution contained 115 mM Na-methanesulfonate, 2.5 mM KOH, 10 mM Hepes, 1.8 mM Ca(OH)$_2$, pH = 7.2.

The online version of this article includes the following source data for figure 6:

**Source data 1.** Conditioning potential and individual rise times related to *Figure 6B*.

**Source data 2.** Prepulse potential and amplitudes normalized to the −20 mV condition related to *Figure 6C*.

decrease in outward current can be observed during the stimulus (see, e.g. traces of E600R in *Figure 1* and traces in *Figure 6*), we never detected in any mutants cumulative inactivation after repeated depolarization, as depicted, e.g., in *Figure 4*. Transition from a conducting to a non-conducting state while the depolarization persists is the definition of inactivation. We nevertheless prefer to refer to the non-conducting states as "closed" rather than "inactivated", because $O_1$ is non-conducting (or not accessed) in the WT, and therefore, the subsequent states cannot be considered inactivated. The state resulting from the flickering block from $O_2$ is arguably inactivated, but such events are commonly not defined as inactivated because of their unstable nature.

In the absence of canonical inactivation, the prepulse-dependent current increase at +40 mV could be related to changes in the relative occupancy of the open states. We hypothesized that the open state $O_1$, which produces larger macroscopic currents, might be more accessible after a hyperpolarization. The current decay after the peak (*Figure 6A*), especially in ΔPASCap, also indicates a transient phenomenon. To map the voltage dependence of this effect more comprehensively, we next compared the effect of hyperpolarized pre-pulse on currents elicited at different test potentials.

We tested the effect of pre-pulse potentials (–160 and –100 mV) on IV protocols in the absence of Cl⁻. Compared to a –100 mV pre-pulse, –160 mV clearly potentiated the first component of ΔPASCap and E600R biphasic IV (*Figure 7A and B*; *Figure 7—source data 1*; *Figure 7—source data 2*).

If the hyperpolarizing potentials facilitate the access to $O_1$ by driving the channel into deep closed states, then impairing the access to these states will reduce the component corresponding to $O_1$ in the GV. The mutation L322H, located in the S4 segment, limits access to deep closed states in WT channels (*Schönherr et al., 1999*). We introduced this mutation in the context of ΔPASCap and E600R. Representative current traces obtained from ΔPASCap$^{L322H}$ and E600R$^{L322H}$ are shown in *Figure 7C and D*; *Figure 7—source data 3*; *Figure 7—source data 4*. Both mutants showed drastic attenuation in the first component of the biphasic GV compared to the parental channels. The tail currents of ΔPASCap$^{L322H}$ and E600R$^{L322H}$ did not show rectification. They presented homogenous kinetics at all potentials, indicating that reducing the access to deep closed states also reduces the occupancy of $O_1$.

## Ca²⁺ calmodulin stabilizes $O_1$

The available cryo-EM structures show $K_V10.1$ in a complex with Ca²⁺-CaM and therefore in a non-conducting state, as it is well established that binding a single Ca²⁺-CaM inhibits the current through WT channels (*Schönherr et al., 2000*; *Ziechner et al., 2006*). In stark contrast, increasing intracellular Ca²⁺ has been reported to potentiate ΔPASCap and E600R current amplitudes (*Lörinczi et al., 2016*). In the light of the results described above, this seemingly paradoxical behavior of mutants could be explained if Ca²⁺-CaM binding restricts the access to the WT-like open state $O_2$ while facilitating access to $O_1$, which gives rise to larger currents and is not visible in WT channels. Therefore, we predicted that Ca²⁺-CaM binding would potentiate the first component of the biphasic IV in the mutants.

To test this, we induced a rise in cytosolic Ca²⁺ using the ionophore ionomycin and inducing release from the stores with thapsigargin (both 5 μM) (*Lörinczi et al., 2016*). Because the increase in intracellular Ca²⁺ is transient and reverts after approximately 150 s (see *Figure 8—figure supplement 1*), and our protocols with discrete voltage pulses require a long time to complete, we used a 5 s voltage ramp from –120 to +100 mV repeated every 30 s for 300 s. The currents were recorded in Cl⁻-free extracellular solution to avoid confounding effects of the endogenous Ca²⁺-dependent Cl⁻ channels. The results of representative experiments on ΔPASCap and E600R are shown in the upper left panels of *Figure 8A and B*, respectively. Sixty seconds after ionomycin/thapsigargin application, a marked potentiation of the current amplitude was observed, and the response to the ramp became linear. The current amplitude declined over the following 300 s, and the biphasic IV characteristics partly recovered. The speed and extent of recovery were higher for ΔPASCap than E600R; after 150 s, the first component recovered by 39.84 ± 6.35% for ΔPASCap and 11.84 ± 3.07% for E600R. This recovery time course matches that of the changes in cytoplasmic Ca²⁺ and agrees with the results obtained by Lörinzi et al. for constant pulses (*Lörinczi et al., 2016*). The magnitude of potentiation and change in IV shape was homogeneous enough among oocytes to allow averaging of the normalized current in all experiments (*Figure 8A and B*, lower panel). The changes in slope can be easily observed in the corresponding first derivative of the normalized IV as a function of voltage shown in *Figure 8—figure supplement 2*.

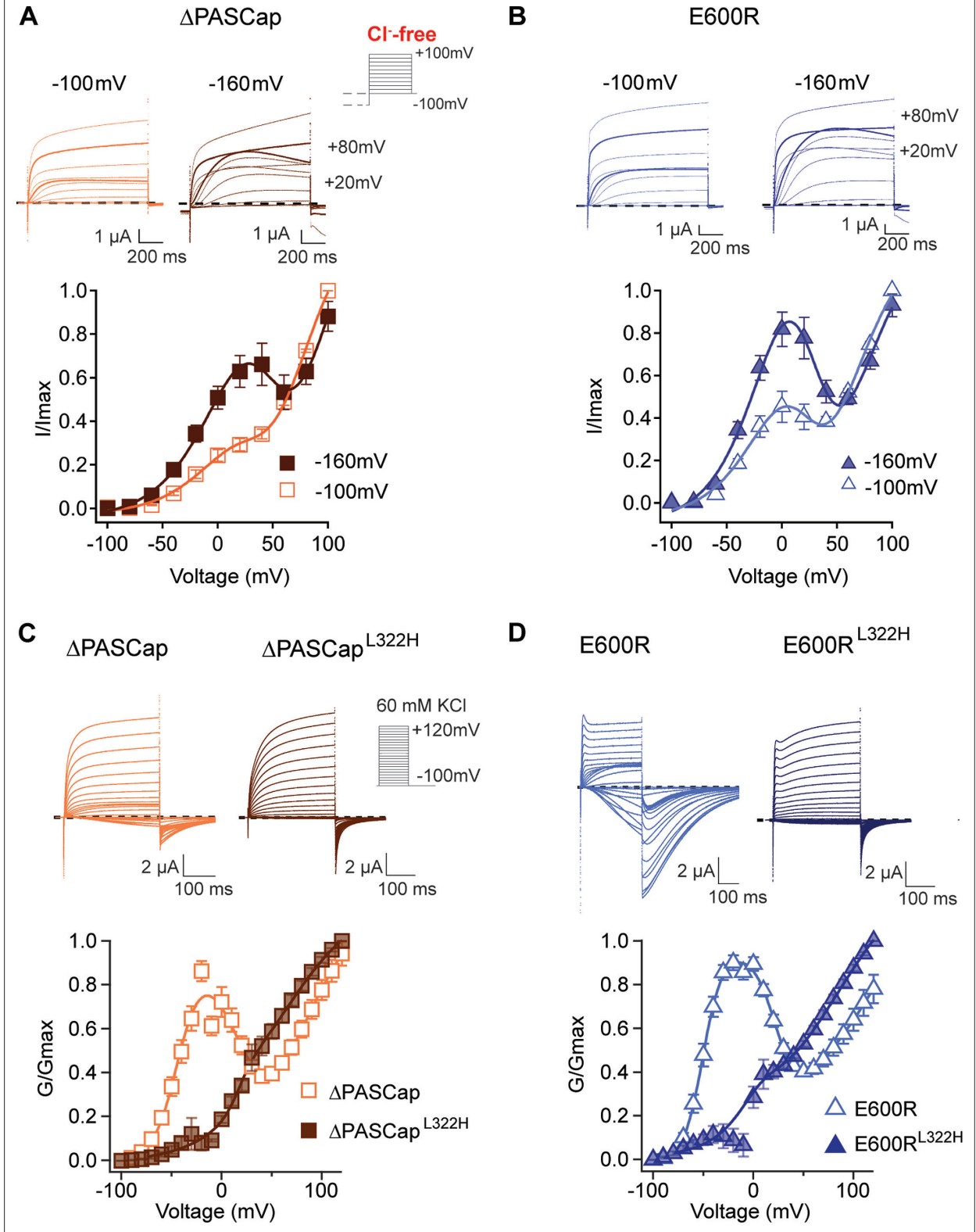

**Figure 7.** Deep closed states facilitate access to $O_1$. (**A, B**) Conditioning pulses to –160 mV potentiated the first component and hence the biphasic behavior of the I/V relationships for ΔPASCap (**A**) and E600R (**B**). (N: ΔPASCap = 7, E600R=6; mean ± SEM) The stimulus protocol is schematically depicted in the inset. A 5 s prepulse to –160 or –100 mV was applied, which is not shown in stimulus cartoons or current traces. The extracellular solution did not contain chloride. (**C, D**) A mutation known to impair access to deep closed states (L322H) largely removes the initial phase of the GV curves

*Figure 7 continued on next page*

*Figure 7 continued*

for ΔPASCap (**C**) and E600R (**D**). (N: ΔPASCap = 7, ΔPASCap$^{L322H}$ = 6, E600$R$=11, E600R$^{L322H}$=7; mean ± SEM). Recordings were performed in a solution containing 60 mM KCl. The stimulus protocols are depicted schematically in the insets. The dashed lines indicate zero current.

The online version of this article includes the following source data for figure 7:

**Source data 1.** Stimulus potential and individual normalized amplitudes in ΔPASCap after prepulse voltages of –100 and –160 mV.

**Source data 2.** Stimulus potential and individual normalized amplitudes in E600R after prepulse voltages of –100 and –160 mV.

**Source data 3.** Stimulus potential and individual normalized amplitudes in ΔPASCap and ΔPASCapL322H.

**Source data 4.** Stimulus potential and individual normalized amplitudes in E600R and E600R L322H.

The changes in amplitude and kinetics in response to rising intracellular Ca$^{2+}$ support our hypothesis that Ca$^{2+}$-CaM stabilizes O$_1$, possibly by driving the channels to deep closed states (***Figures 7 and 8***), which would result in smaller currents in WT and larger currents in mutants with access to O$_1$. We, therefore, predicted that forcing the channels to deep closed states using Ca$^{2+}$-CaM could restore access to O$_1$ in the channels with restricted access to deep closed sates ΔPASCap$^{L322H}$ and E600R$^{L322H}$. This was tested with the approach described above, increasing intracellular Ca$^{2+}$ while recording a repeated ramp protocol in ΔPASCap$^{L322H}$ and E600R$^{L322H}$. As seen in the representative traces in the upper right panels in ***Figure 8A and B***, we observed a notable increase in current 60 s after Ca$^{2+}$ rise, limited to *moderate* potentials, which results in a biphasic behavior resembling that of the ramps in the parental mutants in the absence of Ca$^{2+}$. For both mutants, the IV returned to a linear shape after 300 s. Consistent with the observations for ΔPASCap and E600R, ΔPASCap$^{L322H}$ exhibited faster recovery than E600R$^{L322H}$. Traces normalized to the maximum current and averaged are shown in ***Figure 8A and B*** (lower panel). The respective first derivatives are shown in ***Figure 8—figure supplement 3***.

To further explore the role of CaM, we introduced mutations to preclude CaM binding at the N-terminal (F151N L154N) and C-terminal (F714S F717S) binding sites. We tested the behavior of these mutants (termed BDN and BDC2) in the context of ΔPASCap and E600R constructs (***Figure 8C and D***; ***Figure 8—source data 1***&2) under conditions of basal intracellular Ca$^{2+}$ concentration. ΔPASCap$^{BDN}$ showed a GV relationship like ΔPASCap. In contrast, ΔPASCap$^{BDC2}$ lost its biphasic behavior. In the case of E600R, mutation of the C-terminal binding site (E600R$^{BDC2}$) showed attenuation of the first component of the GV, although not complete. The attenuation of the first component when binding of CaM to its C-terminal binding site is disrupted supports our hypothesis that Ca$^{2+}$-CaM stabilizes O$_1$. The significance of CaM for the mutants seems different. CaM might be crucial for stabilizing O$_1$ in ΔPASCap while less critical in E600R.

These results opened the possibility that the biphasic behavior is due to two coexisting populations of channels, depending on CaM binding at rest. We considered this possibility, but we find it is unlikely because (i) the behavior of the mutants is very homogeneous among oocytes, in which resting Ca$^{2+}$ is very variable; it can be as high as 400 nM (***Busa and Nuccitelli, 1985***), although ten times lower concentrations have also been reported (***Parker et al., 1996***) and (ii) two independent populations of channels would not explain the rectification or the voltage-dependent transitions during short repeated alternating stimuli.

Still, it was possible that CaM is permanently bound to the channel and participates in the gating machinery while only inhibiting the current when bound to Ca$^{2+}$. Although the available literature would not be compatible with this hypothesis (***Schönherr et al., 2000***; ***Ziechner et al., 2006***), we estimated the fraction of channels (WT or mutant) bound to CaM as a function of free Ca$^{2+}$ concentration. We co-expressed Myc-tagged CaM and K$_v$10.1, extracted the proteins in different Ca$^{2+}$ concentrations (see Methods), and pulled down the complex through the Myc-tag of CaM. Under these conditions, we could consistently detect a fraction of channels bound to CaM already in 50 and 100 nM Ca$^{2+}$, which increased dramatically in 0.5 and 1 µM, compatible with previous reports, suggesting that, at basal Ca$^{2+}$, only a small and variable fraction of channels are bound to CaM both in the WT and in the mutants (***Figure 8—figure supplement 4***). The fraction of channels bound to CaM at basal Ca$^{2+}$ (in the range of 4% in 100 nM), even if only one CaM is needed per channel tetramer, is insufficient to explain the prominent and proportionally constant first component observed in the mutants.

In summary, our results strongly suggest that Ca$^{2+}$-CaM stabilizes O$_1$, possibly by driving the channel to deep closed states.

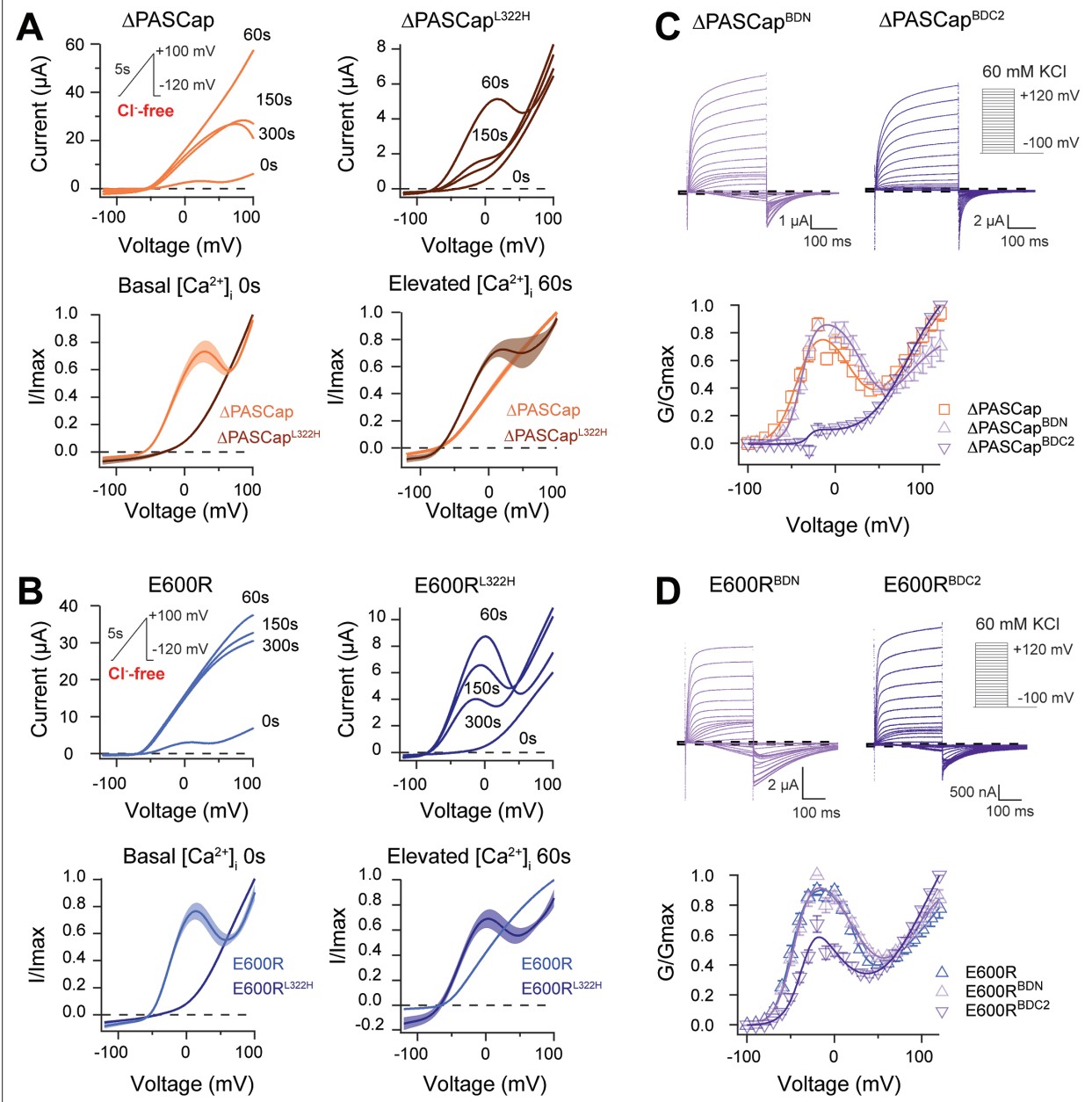

**Figure 8.** CaM stabilizes $O_1$. (**A**) A transient rise in intracellular $Ca^{2+}$ using 5 μM ionomycin and thapsigargin increases ΔPASCap current amplitude (in the absence of external chloride) and the IV relationship becomes linear (upper left traces). The same treatment in a channel carrying a mutation (L322H), reducing access to deep closed states, results in the appearance of a biphasic IV upon $Ca^{2+}$ rise (upper right traces). The lower traces represent the average normalized response comparing 0 and 60 s. (ΔPASCap, light trace, N=8, ΔPASCap[L322H], dark trace, N=5; the shadowed area indicates SEM). (**B**) E600R behavior is comparable to ΔPASCap. Average normalized traces (E600R, light trace, N=10; E600R[L322H], dark trace, N=11; the shadowed area indicates SEM) are represented in the lower panel. The ramp protocol for A and B is schematically represented in the inset. The extracellular solution contained 2.5 mM $K^+$ and no chloride. The dashed lines indicate zero current level. (**C**) Mutation of the C-terminal CaM binding domain (BDC2) in ΔPASCap strongly reduces the first component of the biphasic GV, while deletion of the N-terminal binding site (BDN) did not have any effect (N: ΔPASCap = 7, ΔPASCapB[DN] = 7, ΔPASCap[BDC2]=7; mean ± SEM). (**D**) The reduction of the first component in E600R when the C-terminal CaM binding site is mutated is also present but less intense. (N: E600*R*=11, E600RB[DN] = 9, E600R[BDC2]=11; mean ± SEM). The experimental protocols for C and D are represented in the insets. Currents were recorded in the presence of 60 mM KCl. The dashed lines indicate zero current level.

The online version of this article includes the following source data and figure supplement(s) for figure 8:

**Source data 1.** Stimulus potential and individual normalized amplitudes in ΔPASCap, ΔPASCapBDN ΔPASCapBDC2.

**Source data 2.** Stimulus potential and individual normalized amplitudes in E600R, E600RBDN, and E600RBDC2.

*Figure 8 continued on next page*

**Figure supplement 1.** Chloride currents obtained upon treatment with ionomycin plus thapsigargin.

**Figure supplement 2.** The upper traces show the average of normalized ramps at the indicated times for ΔPASCap (**A**) and E600R (**B**) after induction of $Ca^{2+}$ rise.

**Figure supplement 3.** Like in *Figure 8—figure supplement 2*, the upper traces show the average of normalized ramps for ΔPASCap$^{L322H}$.

**Figure supplement 4.** $Ca^{2+}$ is required for efficient interaction between KV10.1 and CaM.

## A two-layer Markov model recapitulates the kinetic features of ΔPASCap

So far, our experimental results suggest that an additional open state exists in $K_V10.1$ mutants with a compromised intramolecular coupling. This hypothesis can explain the biphasic GV curves, the tail currents' complex shape (*Figures 1 and 2*), the current increase following repeated brief hyperpolarizations (*Figure 4*), and even the paradoxical current increase under rising intracellular $Ca^{2+}$ concentrations (*Figure 8* and its supporting figures). However, in each case, there might be alternative explanations, such as two separate channel populations with different properties or voltage-dependent de-inactivation. The ultimate test for the hypothesis is the construction of a single model that consistently and quantitatively captures all these unusual characteristics.

It should be noted that the model structure presented here (*Figure 9—figure supplement 1*) is not the only one we found to be able to reproduce the data, but it is amongst the simplest that could (see Materials and methods).

Given the large number of model parameters (42+absolute conductance), it might be surprising that all parameters can be constrained. However, the wide range of voltage protocols and the concurrent matching of depolarization and repolarization responses tightly constrains several rate ratios at different voltages and, thereby, ultimately all parameters. This said, the model never fitted all experiments at once very well. Consequently, a global fit proved extremely hard to achieve in an automated way, and we decided to explore the parameter space manually, based on the time constants and rate ratios we could discern from the experiments.

The model parameters are listed in *Supplementary file 2*. These parameters and the structure in *Figure 9—figure supplement 1* jointly define the model that produces all results in *Figure 9* and *Figure 9—figure supplements 1–5* for ΔPASCap.

Modifications are only necessary to capture the effect of $Ca^{2+}$-CaM-binding and the properties of other channel variants. Suppose the binding influences the interaction between the intracellular modules that stabilize different configurations of the gating ring, e.g., BDC2, CNBHD, BDC1, and PASCap. In that case, the free energy of these states will change upon $Ca^{2+}$-CaM binding. Consequently, the free energy difference between the states, and thereby the apparent voltage sensitivity of the transitions, will change too. Hence, we represented the effect of different degrees of $Ca^{2+}$-CaM binding as different voltage shifts in $\lambda$, $\kappa_L$, $\gamma$, as indicated in *Figure 9E*.

While the Markov model describes the time and voltage dependence of state occupancies, the experimental observable is current. From the model's occupancies, current was obtained by application of the GHK flux equation (*Equation 4*) for both open states. While the single channel conductance of the two open states in the model is identical, the effective occupancy of $O_2$ and, thereby, its macroscopic conductance is drastically reduced by a flicker block, a rapid transition into $C_2$, and a slower return to $O_2$. To match experimental results, in particular the pre-pulse-dependent activation and tail current amplitudes, the flickering kinetics had to be adjusted to achieve a limiting occupancy ratio of 5.5 between $C_2$ and $O_2$, i.e. $r_{close}/r_{open}=4.5$. Typically, flicker-closures are thought to result from instabilities of the open state, unrelated to voltage sensor rearrangements. Here, $r_{close}$ and $r_{open}$ are constant and voltage-independent. The range of plausible absolute values for $r_{close}$ and $r_{open}$ is well constrained by experiments. If, following strong depolarizations, $O_2$ is populated faster than it is vacating into $C_2$, a transient overshoot appears shortly after depolarization onset. This is observed in E600R, but not ΔPASCap (*Figure 1A and C*). We found that the E600R observations can be mimicked with $r_{open} = 30$, while mimicking ΔPASCap activation requires at least $r_{open} = 100$. Further increases induce only small changes, visible in the triphasic tail current kinetics. In the limit of very fast flickering kinetics, the model becomes indistinguishable from a model without flickering in which $O_2$ microscopic conductance is reduced to 1/5.5.

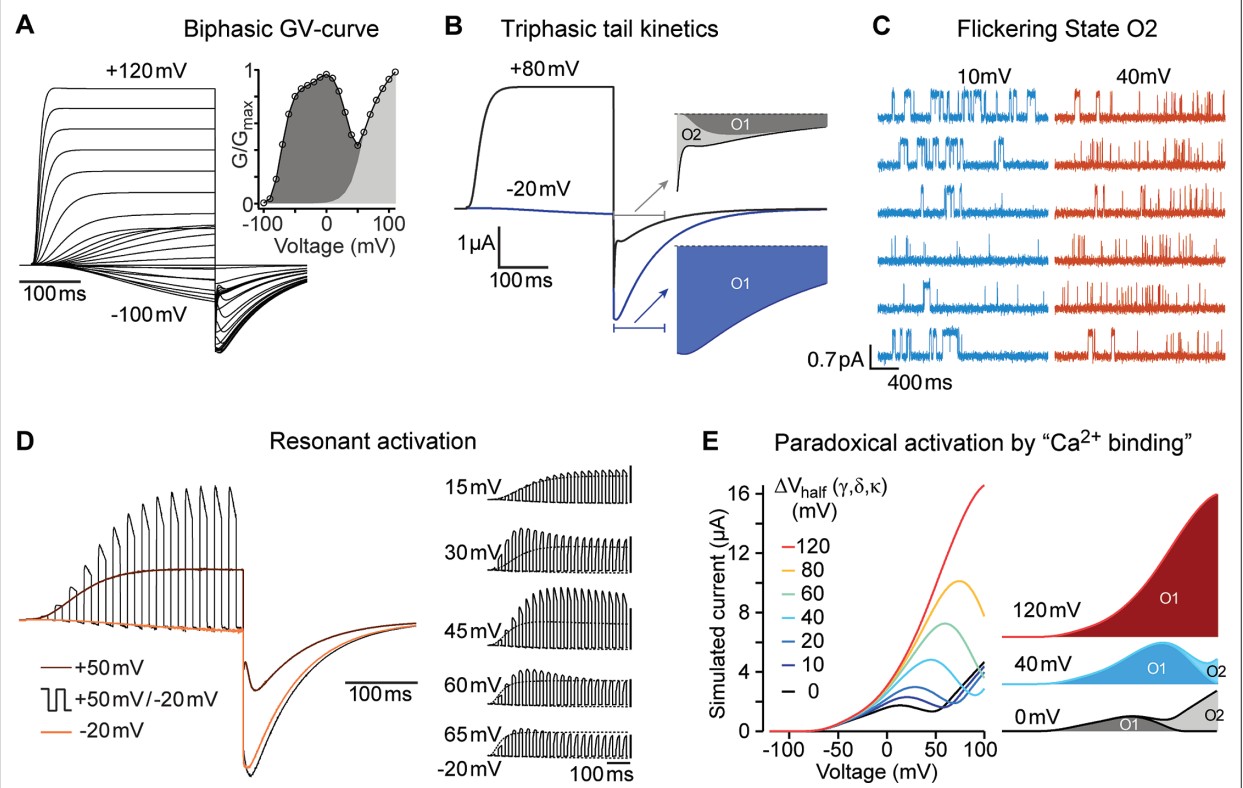

**Figure 9.** A single model can reproduce all experimental observations. (**A**) In response to an I-V-protocol (**Figure 1A**), the model displays biphasic activation, more clearly represented in the derived G-V curve (right, compare to **Figure 1B**). The filled area indicates the contributions of the two open states. The tail-currents show a complex dependence on test-pulse voltage and time. (**B**) Two traces from A, shown with longer repolarization. The first 100ms of the tail currents are displayed on an extended timescale. Colored areas indicate the contributions of the two open states. After weak depolarization to –20 mV, the tail-current originates almost entirely from the mutant-specific open state $O_1$ (blue area). After strong depolarization, $O_2$ mediated currents initially dominate (light grey). For details, see **Figure 9—figure supplement 2**. (**C**) The model predicts the differences in single channel behavior of ΔPASCap upon depolarization to +10 or+40 mV (Compare to **Figure 5**). (**D**) In response to 10ms pulses alternating between –20 and +50 mV, the model shows currents that exceed the currents obtained with constant pulses to +50 mV (compare **Figure 4B**). Relating the currents during the positive and negative pulses to the concurrent currents elicited by the two constant pulses (brown and orange), the ratio lies by about 1.6 for the period starting 200ms after the pulse onset. This excess current depends sensitively on the duration and voltage of the two pulse components. The right series of simulations displays the results for 15ms pulses to –20 mV, alternating with 15ms pulses to voltages from 15 mV to 65 mV. The corresponding responses to the constant pulses are displayed with thin dotted lines. To facilitate perception of the excess current, the five groups of traces are scaled individually, so that the peak amplitude of the dotted response elicited by the stronger depolarizations is displayed at equal size throughout. The vertical scalebars correspond to the same absolute current. From top to bottom, the excess current ratios at 200ms changes are 1.05, 1.38, 1.74, 1.46, and 1.16. (**E**). The binding of $Ca^{2+}$-CaM is implemented through change in the activation energy, corresponding to a shift in the equilibrium voltage of the gating transitions. The decomposition of the current into the individual open states' contribution shows that for increasing voltage shifts – representing high $[Ca^{2+}]_i$ – the mutant-specific $O_1$ closes later into the ramp, until eventually all current is carried by $O_1$.

The online version of this article includes the following figure supplement(s) for figure 9:

**Figure supplement 1.** Cartoon depicting the state model proposed.

**Figure supplement 2.** The triphasic tail currents result from transient re-population of $O_1$ during channel deactivation.

**Figure supplement 3.** Current simulations for ΔPASCap upon depolarization to +20 and+80 mV after a conditioning pulse to –160 or –100 mV.

**Figure supplement 4.** Side-by-side comparison of experimental data and model prediction during depolarizations between –100 and +100 mV in the presence of high (60 mM) (**A**) or low extracellular [K+] (2.5 mM) (**B**).

**Figure supplement 5.** Model variants can account for features of related channels and mutated variants.

To test the model, we focused on the most conspicuous kinetics features observed with ΔPASCap. The first feature was the tail kinetics. In contrast to the slow monophasic deactivation observed in response to *weak* depolarizing pulses (–20 mV), triphasic tail kinetics was detected in response to *strong* depolarizing pulses (+80 mV; **Figure 1C**). The model could replicate the slow deactivation

after *weak* depolarizations, fast after *strong* depolarizations, and mixed kinetics on *moderate* stimuli (*Figure 9B*; *Figure 9—figure supplement 2*).

The model also reproduces the effect of a hyperpolarizing prepulse on different test potentials. As described (*Figure 6*), $O_1$ is preferentially accessed from deep closed states, which correspond to states towards the left in the lower layer in the model. It is plausible that a hyperpolarizing pre-pulse (–160 mV) drives the channel to occupy the deep closed states (lower layer), while a pre-pulse of –100 mV distributes the channel between both layers. We, therefore, adjusted the parameters accordingly (*Supplementary file 2*) and simulated the current trace (*Figure 9—figure supplement 3*). We focused on a test pulse that represents *moderate* depolarizations (+20 mV), and compared it to a *strong* depolarizing pulse (+80 mV). Like in the experiment, the more strongly hyperpolarizing prepulse (–160 mV) potentiates the current at +20 mV, as compared to the prepulse to –100 mV (*Figure 9—figure supplement 3A*). This does not happen for the larger depolarization to +80 mV (*Figure 9—figure supplement 3B*).

The model also recapitulates the behavior of ΔPASCap during repeated short stimuli between –20 and +50 mV. The current amplitude driven by intermittent stimuli is larger than that driven by a sustained stimulus to the same potential. The model also reproduces the relative size and the kinetics of the tail current (*Figure 9C*; *Figure 9—figure supplement 4*).

The effect of a rise in $Ca^{2+}$ can be reproduced by shifting the voltage dependence of just two transitions (γ↔δ and κ). The rationale behind this that $Ca^{2+}$-CaM binding stabilizes the basal config-uration of the ring and thereby shifts the activation energy for state transitions that are coupled to ring reconfigurations, i.e. the second VSD displacements (γ↔δ) and the ring gating itself (κ). Shifts in activation energy of voltage-dependent rates correspond to an extra term that can be expressed as voltage offset. When the voltage-dependence of these three rate constants is shifted by 10–120 mV, the occupancy of states in the "lower" layer is increased and thereby the access to $O_1$ (*Figure 9D*). The resulting simulated currents reproduce the experimental observations: a linear current response during ramps in high intracellular $Ca^{2+}$.

To describe the behavior of $K_V10.1$ WT, we only needed to remove the access to $O_1$ and shift the parameters of the rotation of the ring in the hyperpolarizing direction to reflect the more stable struc-ture resulting from intact interactions among intracellular domains and between these and the core of the channel. The model recapitulated the change in kinetics depending on the pre-pulse potential.

The model structure and the values of its parameters have been developed to reproduce the kinetic features of currents through $K_V10.1$ ΔPASCap channels. Can it provide insights into specific properties of other genotypes? A very palpable, E600R-specific feature is the transient current peak that occurs within about 30ms after depolarizations above +70 mV (*Figure 1A and C*). It suggests that $O_2$, the open state underlying current at these voltages, is populated and vacated with very similar rates. In the model, such a transient peak can indeed be reproduced if the rate constant from $O_2$ into $C_2$ (*Figure 9—figure supplement 1*) is reduced from 100 $s^{-1}$ for ΔPASCap to 30 $s^{-1}$ for E600R (*Figure 9—figure supplement 5*). E600R also supports larger tail currents, relative to the test-pulse currents, indicating an even stronger bias away from $O_2$ into $C_2$. To replicate this, $r_{close}/r_{open}$ was increased to 11 and the rate constant ζ, leading out of $O_2$ was adjusted to compensate for the lower $O_2$ occupancy. These few changes, all concerned with $O_2$ instability, underlie the simulations in *Figure 9—figure supplement 5* A and B. The corresponding parameters are given in *Supplementary file 2*.

The tails observed in E600R show a shape reminiscent of that of $K_V11.1$, where slow deactivation is crucial for its physiological function. Our model can mimic $K_V11.1$ if the flicker block is biased further towards the closed state, and the transitions between closed states are slowed (*Figure 9—figure supplement 5* D and E; *Supplementary file 2*).

## Discussion

The wide diversity of electrical responses in cells relies greatly on subtle differences in the behavior of voltage-gated channels. Despite the many relevant advances in the knowledge of the structure of ion channels, the correlation between the structures and the functional determinants of channel behavior is incompletely understood. In this study, we have combined biophysical, biochemical, and mathematical approaches to understand the complex gating behavior of $K_V10.1$ potassium channels, which is the basis of a group of diseases with devastating consequences (e.g. *Toplak et al., 2022*; *Tian et al., 2023*).

In *KCNH* channels, intracellular domains contribute to gating by forming an intracellular ring that rotates in response to depolarizing stimuli (***Mandala and MacKinnon, 2022***). We have studied a series of mutant channels where the integrity of the intracellular ring is compromised by either deletions or point mutations. In all the mutants, the G-V relationship shows a biphasic behavior with evident inward rectification at intermediate depolarizations and a complex deactivation with kinetics also dependent on the stimulus potential (***Figure 1***). Our observations can be explained by the coexistence of two different open states, one of which corresponds to the open state observed in WT ($O_2$), while the other one ($O_1$) is made visible as a result of the modification of the intracellular gating ring. We hypothesize that $O_1$ corresponds in the WT to the 'non-conducting state 2' identified in the closely related $K_V10.2$ channel by cryo-EM (***Zhang et al., 2023***), in which flipping of Y460 (Y464 in $K_V10.1$) renders a hydrophobic constriction wider than 8 Å, enough to allow $K^+$ flow, but still corresponds to a non-conducting state. The presence of an intact intracellular ring would preclude ionic flow in the WT, and its alteration would explain the permeability of this state in the mutants.

The coexistence of two independent channel populations with different gating properties could be an alternative explanation for the observed behavior. Still, it is very unlikely that two such populations would be expressed at the same ratio in all oocytes, given the variability of the system, and the behavior of each of the mutants was highly reproducible among oocytes obtained from different frogs and time points. Furthermore, activity in excised patches from oocytes expressing mutant channels was abolished in the presence of astemizole.

The atypical open state $O_1$ allows the study of intermediate steps between the two major gating events, VSD displacement and ring rotation. Upon intermediate depolarization, the channels would access the open states sequentially, while $O_2$ is predominant on strong depolarizing steps and $O_1$ upon mild stimuli. The analysis of the GV of the mutants using global parameters (***Figure 2***) revealed that they all share the component responding to mild depolarizations ($O_1$) and only the second component ($O_2$) and (probably most importantly) the transition between components depends on the particular mutant studied. The larger the deletion in the intracellular ring is, the stronger the shift in the voltage dependence of the second component. Importantly, this observation is also incompatible with two independent populations of channels. Since the two gating steps are sequential (***Mandala and MacKinnon, 2022***), the displacement of the VSD remains the main factor governing the speed and voltage dependence of the activation. Thus, changes in the extracellular $Mg^{2+}$ concentration, which are known to interfere with the movement of the VSD (***Bannister et al., 2005***; ***Silverman et al., 2004***), cause a shift of the voltage dependence and activation speed that affects both gating components (***Figure 3***), but not the transition between them.

The biphasic behavior arises from a different apparent conductance between the two open states, with larger open probability and, therefore, larger macroscopic amplitude for $O_1$. This results in a decrease in current amplitude as the channels leave $O_1$ and transition to $O_2$. Since the time required to access the two states is different, when short alternating stimuli are applied, each with a too short duration to allow entry into $O_2$, the longer open time of this state results in a larger current amplitude than when a single sustained stimulus is used (***Figure 4***). The presence of a second state with larger or longer openings when the integrity of the ring is compromised could explain the larger current amplitudes observed in heteromeric $K_V11.1$ (HERG1a/1b) channels, which lack at least one PAS domain as a result of alternative splicing and are crucial for proper cardiac repolarization (***Feng et al., 2021***). Thus, the second open state could also have physiological relevance in naturally occurring channel complexes.

Access to $O_1$ is favored from deep deactivated states, which appear important in other aspects of $K_V10.1$ channel function (e.g. Cole-Moore shift). Our conclusion is based on the changes in amplitude observed with different pre-pulse potentials (that is, driving the population to deep closed states, ***Figure 7A and B***) and on the behavior of a mutant known to hinder access to such deactivated states (L322H), which, when combined with mutations revealing $O_1$, shows limited access to this state (***Figure 7C and D***).

$Ca^{2+}$-CaM is an important modulator of $K_V10.1$ that reduces WT current amplitude in the presence of elevated $Ca^{2+}$ levels. A paradoxical current increase had been described for some of the mutants used in this study (***Lörinczi et al., 2016***), and we speculated that the presence of $O_1$ could be the basis for this phenomenon. Indeed, the elevation of intracellular $Ca^{2+}$ leads to a transient loss of the biphasic behavior and a larger current amplitude of the mutants compatible with a larger fraction of

channels in $O_1$. Because access to $O_1$ occurs from deep closed states, this could be explained by an increased occupancy of such deactivated states in response to CaM binding. This appears to be the case since CaM induces a biphasic behavior in the mutant channels that show reduced access to deep closed states; thus, L322H mutants behave like the parental variants in the presence of $Ca^{2+}$-CaM. This implies a mechanistic explanation for the effect of $Ca^{2+}$-CaM on WT since favoring entry into deep closed states would result in a decrease in current amplitude in the absence of (a permeable) $O_1$.

Our initial hypothesis that CaM participates constitutively in the gating machinery of the channel, based on the loss of biphasic behavior when the C-terminal binding site for CaM was mutated (*Figure 8D*), is unlikely to be correct because although there is a significant binding of CaM to the channel at basal intracellular $Ca^{2+}$, this fraction of channels, combined with the strong increase in bound CaM in the presence of high $Ca^{2+}$ and the variable intracellular basal $Ca^{2+}$ in oocytes would be insufficient to explain the qualitatively consistent behavior of the current of the different mutants. We speculate that the effects of the mutations in CaM binding sites are more related to their location in the protein than to their ability to bind CaM.

In summary, the gating of $K_V10.1$ (and similar channels) consists of a sequence of events that affect the voltage sensing domain, which moves in two sequential steps (*Schönherr et al., 1999*) and whose movement is transferred to the pore domain through intramolecular interactions (*Bassetto et al., 2023*). The voltage sensor maintains the gate closed (*Mandala and MacKinnon, 2022*), and its displacement has a permissive effect on the gate opening. Once this restrictive factor is removed, a second step occurs, most likely corresponding to a rotation of the intracellular ring that finally allows the gate to relax to the open state (*Patlak, 1999*). This final step is the only one directly observed in WT channels.

We proposed the existence of a second open state in $K_V10.1$ with an altered eag domain, because that could readily explain otherwise puzzling observations in macroscopic recordings. Our single channel recordings provided definitive evidence for channel openings with two different kinetic characteristics. Finally, we constructed a Markov model of $K_V10.1$ mutants' gating that replicates all key experimental observations (*Figure 9* and its supplements). This model performance required two modifications to previously established $K_V10$ models: 1. introduction of an additional open state with a larger macroscopic current and 2. an additional gating step orthogonal to the previously described state plane. This added a second layer of states. Steps within each layer correspond most likely to two sequential displacements of the four VSD. Steps between the layers are interpreted as reconfiguration of the intracellular ring. The new state $O_1$ is already accessible when the first displacement of all four VSD is completed but the ring has not changed, while entry into $O_2$ requires complete VSD displacements and ring reconfiguration. The rapidly flickering open state of $O_2$ observed in the single channel recordings, is realized in the model by rapid, voltage-independent gating in-and-out of a neighboring closed state ($C_2$). This flicker block might also offer an explanation for a feature of the mutant channels, that is not explained in the current model version: the continued increase in current amplitude, hundreds of milliseconds into a strong depolarization (Supp. 4 to *Figure 9*). If the relative stability of $O_2$ and $C_2$ continued to change throughout depolarization, such a current creep-up could be reproduced. However, this would require either the introduction of further layers of $O_n \leftrightarrow C_n$ states, or a non-Markovian modification of the model's time evolution.

To which degree does the model reflect the topology and gating transitions of $K_V10.1$ mutants? By construction, the presented model has one of the simplest structures that can replicate all key findings. Specifically, a model with less than three different gating steps (two VSD, one ring), without flicker block, or with $O_1$ and $O_2$ in close proximity would fail to replicate at least one of the key findings. With respect to these fundamental properties, the model's structure should mirror the fundamental properties of $K_V10.1$ gating. The finer details of the model structure are less certain; we were able to vary them without large improvements in performance, but also without clear losses. This includes the feature that trans-layer connections exist for each state except the open states and the flicker block, and the feature of complete equivalence of the VSD transitions within lower and upper layer. Those features were made as choices for simplicity and symmetry. One price for the simple structure is the relative complex voltage dependence of the rates. The functional form of the rates' voltage dependence is an often-underappreciated way to shift model complexity from the state diagram to the rate functions. Following Arrhenius or Eyring formalisms, transition rates are expected have the form $c_0 \cdot \exp(\Delta V/k)$ and range from zero to infinity without saturation or a non-zero minimal rate. If, however,

a transition between two states ($\leftrightarrow S_1 \leftrightarrow S_2 \leftrightarrow$) is used as a shorthand for an underlying linear three-state system ($\leftrightarrow S_1 \leftrightarrow S_2 \leftrightarrow S_3 \leftrightarrow$), in which only the transition $S_1 \leftrightarrow S_2$ is voltage-dependent, while $S_2 \leftrightarrow S_3$ features constant forward and backward rates, then the current activation kinetics in the two-state shorthand will saturate even for infinitely high voltages. Similarly, a non-zero basal rate $x_0$ can be caused by an alternate 'leakage' route from the first state $S_1$ to the open state O that occurs at a very low, voltage-independent rate. In the light of such considerations, the sigmoidal voltage dependences of the model's rates point to a much more complicated underlying structure.

A large majority of ion channel models in use today uses voltage dependent rates with functional forms that are more complex than a simple exponential. Although in general, we prefer the use of the simple forms (e.g. *Yang et al., 2016*), in the present case, we decided to keep the model structure as simple as possible and use a more complex voltage dependence of the rates, to avoid adding even more states. As a consequence, some voltage dependencies can raise questions about how the model corresponds mechanistically to underlying gating steps. In particular rate κ, governing the transition from the newly introduced upper floor to the ground floor, is described by a sum of two voltage dependencies of opposite sign. The necessity for such a complex voltage dependence clearly signals unresolved structural details. What's more, the κ/$\lambda$ transition could reasonably be expected to be voltage independent because we related it to ring reconfiguration, a process that should occur as a consequence of a prior VSD transition. We have made some attempts to treat this transition as voltage independent but state-specific with upper-layer bias for states on the right and lower-layer bias for states on the left. This is in principle possible, as can already be gleaned from the similar voltage ranges of the left-right transition (α/β) and the $\kappa_L$/$\lambda$ transition. However, this approach leads to a much larger number of free, less well constrained kinetic parameters and drastically complicated the parameter search. The model version presented here is therefore a low-complexity compromise that showcases the explanatory power of a second open state and an additional gating step.In summary, this study presents a more complete description of the gating mechanism of $K_V$10.1 channel, which, importantly, can be extended to other members of the *KCNH* family. In response to depolarization, the movement of the voltage sensor would have a permissive role in the opening of the gate. The rotation of the intracellular ring would be the effective unlocking mechanism allowing permeation. This has profound implications pertaining to the possibilities of fine-tuning gating since the intracellular ring is more susceptible to posttranslational modifications and protein-protein interactions than the transmembrane domains. Our current knowledge of the physiology and pathophysiology of $K_V$10.1 indicates that the channel is relevant for the regulation of excitability acting at potentials close to the resting rather than during active electrical signaling. Therefore, sustained modulation of gating, possible through modification of the intracellular ring, would be crucial for channel function. Since they allow dissection of the ring-dependent effect, our mutants will allow for a direct study of such modulation mechanisms.

## Materials and methods
### Constructs
Mutants were generated using $K_V$10.1 (hEAG1) cloned in pSGEM (M. Hollmann, Bochum University) as a template (*Jenke et al., 2003*). The deletion mutants, Δ2–10 and ΔPASCap, were generated using In-Fusion HD Cloning kit (Clontech (TaKaRA)) following the supplier's instructions. For each construct, we designed two primers, each of them with two regions: a 3' region that anneals to the template immediately up- or downstream of the sequence to be deleted and a 5' that does not bind to the template but overlaps with the second primer (*Supplementary file 3*). The subsequent PCR amplification will then omit the sequence between the hybridization sites for the primers.

To generate the point mutations (ΔPASCap[L322H], ΔPASCap[BDN], ΔPASCap[BDC2], E600R[L322H], E600R[BDN], E600R[BDC2]), site-directed mutagenesis was performed using Quick Change II XL kit (Agilent Technologies) using the primers listed in *Supplementary file 3*. E600R.pLeics71 and Δeag.pLeics71 were a gift from Dr. John Mitcheson, University of Leicester.

The expression construct for 5-myc-calmodulin was obtained by restriction cloning of CALM1 from pKK233-hCaM, which was a gift from Emanuel Strehler (Addgene plasmid # 47598) (*Rhyner et al., 1992*) into a pSGEM construct with five consecutive repeats of the myc tag (*Lörinczi et al., 2015*) using NcoI and HindIII (New England Biolabs).

All plasmids were linearized with NheI, and cRNA was synthesized using mMESSAGE mMACHINE T7 Transcription kit (Invitrogen Ambion).

## Two-electrode voltage-clamp recordings

Oocyte preparation and injection were performed as described (*Stühmer, 1992*). The amount of cRNA injected depended on the current amplitude obtained with each construct: 0.075–0.5 ng/oocyte (WT and point mutation), 0.075–5 ng/oocyte (deletion mutants), and 8–10 ng/oocyte (split channels). Oocytes were maintained at 18 °C in ND96 buffer (in mM: 96 NaCl, 2 KCl, 1.8 CaCl$_2$, 1 MgCl$_2$, 5 HEPES, 2.5 Na-pyruvate, 0.5 Theophylline, pH 7.55). Tetracycline (USB) (50 µg/ml), Amikacin (Enzo) (100 µg/ml), and Ciprofloxacin (Enzo) (100 µg/ml) were added as recommended in *O'Connell et al., 2011*.

Two-electrode voltage-clamp (TEVC) recordings were performed 1–5 days after oocyte injection. The intracellular electrodes had a resistance of 0.4–1.5 MΩ when filled with 2 M KCl. Normal Frog Ringer solution (NFR): (in mM: 115 NaCl, 2.5 KCl, 10 HEPES, 1.8 CaCl$_2$, pH 7.2), was used as external solution. Higher K$^+$ concentration (mM: 60 KCl, 57.5 NaCl, 10 HEPES, 1.8 CaCl$_2$, pH 7.4), was used instead to examine tail currents. Cl$^-$-free NFR (in mM: 115 Na-methanesulfonate, 2.5 KOH, 10 HEPES, 1.8 Ca(OH)$_2$, pH 7.2), was used to limit current contamination with Cl$^-$ current; in this case, agar bridges (2% agar in 3 M NaCl) were used for the reference electrodes. To raise the intracellular Ca$^{2+}$ concentration (*Lörinczi et al., 2016*), 5 µM ionomycin (Abcam) and 5 µM Thapsigargin (Abcam) were added to the bath. Ionomycin and Thapsigargin 5 mM stocks were prepared using DMSO and diluted in the recording medium (Cl$^-$-free NFR) immediately before recording. The final concentration of DMSO was thus 0.1%.

Data acquisition was performed using a TurboTEC 10 CD amplifier (npi Electronics) and the ITC-16 interface of an EPC9 patch-clamp amplifier (HEKA Elektronik). The current was filtered at 1.3 KHz and sampled at 10 KHz. Patchmaster software (HEKA Elektronik) was used to design and apply the stimulus protocols applied. Because of the profound effects of hyperpolarization on channel kinetics, leak subtraction was avoided except when explicitly indicated. Linear leak was subtracted offline assuming a reversal potential of 0 mV. Fitmaster (HEKA Elektronik) and IgorPro (WaveMetrics) were then used to analyze the recordings.

Most conductance-voltage plots were obtained for recordings with [K$^+$]$_{ext}$ =60 mM. In this condition, tail currents are large and allow precise estimation of the reversal potential $V_{eq}$. Under this condition, the difference between intra- and extracellular potassium concentration is small and the Goldman-Hodgkin-Katz flux equation predicts a nearly linear relation. In these cases, overall conductance was calculated from the end-pulse current after measuring the reversal potential ($V_{eq}$) using:

$$G = I / \left( V_m - V_{eq} \right) \tag{1}$$

where I is the current amplitude and $V_m$ the stimulus. Reversal potential was calculated using the end pulse current for mutant channels activating below the equilibrium potential for potassium, or instantaneous IV using tail currents for channels opening at more depolarized potentials. Conductance was then normalized to the maximum value and plotted against voltage stimulus. For WT recordings (*Figure 1B*), a sigmoidal response was then fitted with a Boltzmann equation:

$$f \left( V \right) = \left( 1 + \exp \left( \left( V_h - V_m \right) / K \right) \right)^{-1} \tag{2}$$

With $V_h$ being the voltage for half-maximal activation, K the slope factor and $V_m$ the membrane potential as above.

The biphasic response we observed in the mutants was described with two sigmoidal components and a weight W to represent the transition between the two components. The equation used was as follows:

$$f \left( V \right) = A_o + \left( 1 - W \left( V \right) \right) \cdot A_1 \cdot \left( 1 + e^{\left( V_{h1} - V_m \right) / K_1} \right)^{-1} + W \left( V \right) \cdot A_2 \cdot \left( 1 + e^{\left( V_{h2} - V_m \right) / K_2} \right)^{-1} \tag{3a}$$

$$W \left( V \right) = \left( 1 + e^{\left( V_{h3} - V_m \right) / K_3} \right)^{-1} \tag{3b}$$

If the currents were recorded in $[K^+]_{ext}$=2.5 mM, the driving force for currents becomes considerably non-linear (*Kotler et al., 2022*) and we estimated the conductance based on the full Goldman-Hodgkin-Katz flux equation for a current surface density $\Phi$.

$$\Phi = P \cdot P^{open} \cdot zF \cdot \frac{V_m}{V_T} \left( \frac{[K^+]_{in} - [K^+]_{out} \cdot e^{-V_m/V_T}}{1 - e^{-V_m/V_T}} \right),$$

where $V_T = \frac{RT}{zF}$, with the gas constant $R$, the Faraday constant $F$, the absolute temperature $T$, the potassium channel open probability $P^{open}$, and the maximum permeability $P$. The ion's valence $z$ equals 1. Because we only work with normalized conductances, the following relation can be used to determine the voltage-dependent conductance:

$$G(V) \propto I \cdot \left( \frac{1 - e^{-V_m/V_T}}{[K^+]_{in} - [K^+]_{out} e^{-V_m/V_T}} \right) \cdot \frac{V_T}{zFV_m} \qquad (4)$$

## Single-channel recording and analysis

The vitelline membrane of oocytes expressing ΔPASCap cRNA, as described for two-electrode experiments, was mechanically removed after osmotic shock (*Stühmer, 1992*) to expose the plasma membrane. Patch pipettes were pulled from thick-wall (0.5 mm) borosilicate capillaries to obtain a resistance of 7–12 MΩ when filled with the intracellular solution, which contained (mM) 100 KCl, 10 EGTA, 10 HEPES pH 7.2 (KOH). Recordings on outside-out patches were acquired using an EPC10 Plus amplifier and PatchMaster software (HEKA Elektronik). Currents were digitized at 10 kHz and filtered at 2 kHz. Analysis was performed using FitMaster. For presentation, the traces were digitally filtered at 100 Hz.

## Pull-down

*Xenopus laevis* oocytes were co-injected with RNA coding for 5xmyc-calmodulin (10 ng) and $K_V$10.1 (0.5 ng). Oocytes were lysed 72 h after injection through mechanical disruption in lysis buffer and incubation for 30 min on ice. The lysis buffer (1% Triton X-100, 150 mM NaCl, 50 mM HEPES pH 7.4, cOmplete EDTA-free protease inhibitor cocktail (Roche)) contained different free $Ca^{2+}$ concentrations (0, 20, 50, 100, 500, and 1000 nM). To obtain accurate $Ca^{2+}$ concentrations, EGTA was titrated with $CaCal_2$, to prepare 100 mM CaEGTA stock solution as described in *Tsien and Pozzan, 1989*. CaEGTA and EGTA were then added to the lysis buffer, adjusting EGTA/CaEGTA to control the concentration of free $Ca^{2+}$ taking into account pH, ionic strength and temperature of solution using MaxChelator (*Bers et al., 2010*). As controls, non-injected oocytes, and oocytes injected with only $K_V$10.1 or 5xmyc-calmodulin were lysed in a $Ca^{2+}$ free buffer.

The lysate was then centrifuged (20,000 x*g*) at 4 °C for 3 min to remove debris. 10% of the supernatant was set aside to load on the gel as input control. The lysate was pre-cleared using protein G magnetic beads (New England Biolabs). The cleared supernatant was then incubated with 3 µg anti-myc antibody (SIGMA M4439 monoclonal anti c-myc or Abcam ab206486 rat mAb to myc tag (9E10)) for 1.5 hr at 4 °C. Protein G magnetic beads were then added and incubated for 1.5 hr at 4 °C in rotatio beads were washed three times using 0.1% Triton X-100, 300 mM NaCl, 50 mM HEPES pH 7.4, cOmplete (EDTA-free protease inhibitor cocktail), EGTA and CaEGTA to obtain the corresponding $Ca^{2+}$-free concentrations (see above). Electrophoresis and immunoblotting conditions were as previously described (*Lörinczi et al., 2015*) using an anti-Myc (Sigma, 1:1000) or an anti-$K_V$10.1 antibody (*Chen et al., 2011*) overnight.

## Markov-model construction

The standard model for $K_V$10.1 gating comprises two gating steps for each of the four subunits' voltage sensors. The first step unfolds in the hyperpolarized voltage range and forms the basis of the Cole-Moore shift, characteristic of $K_V$10.1 channel gating. The second, faster step readies the channel for opening, and once it is performed in all subunits, a conducting state can be reached (see also *Mandala and MacKinnon, 2022*). While this model captures the key features of WT activation, we had to extend it to account for the features of mutant gating. Throughout the model development and tuning, we focused on the experiments performed with the ΔPASCap mutant. However, a limited

number of tests with other parameters showed that experimental findings with E600R and even with $K_V$11.1 can also be matched (Supp.5 to *Figure 9*).

In a first step, we attempted a minimal model extension by attaching an additional open state to some closed state of the standard model. None of the resulting models, for any choice of attachment site and rate voltage dependence, could replicate the multi-phasic tail currents, the pre-pulse dependent activation, and the delayed opening. We came to this conclusion in two steps. First, we could exclude many possible attachment sites because the attachment site must be sufficiently far away from the conventional open state. Otherwise, the transition from 'O$_1$ preferred' to 'O$_2$ preferred' via a few closed intermediate states is very gradual and never produces the biphasic GV curves. Second, we found that the first gating transition in the standard model (along α, left to right) can either produce a sigmoidal current onset or bias the model for occupancy of state O$_1$ over O$_2$, but not both. Waveforms such as the dark orange trace in *Figure 6A*, in response to a –160 mV pre-pulse and a 20 mV test pulse require both a bias towards O$_1$ and a sigmoidal onset. We found that this can only be accounted for by introducing a third gating step orthogonal to the two transitions in the standard model.

Without a priori information about the new states introduced in this way, we decided to simply add a copy of the standard model as an additional layer, an 'upper floor'. Crossing from the ground floor to the upper floor was made possible for any state. From a structural view, this newly added transition might be related to a reconfiguration of the gating ring. Considerations as given above led us to attach the mutant-specific O$_1$ to the basal level and the conventional O$_2$ to the upper level. The experimental data indicated that O$_1$ had to be accessible to states 'on the right of the ground floor', that is, accessible from deep closed states. Along the same lines, O$_1$ could not be located 'too far toward the bottom' of the scheme, because it is an open state. Eventually, we decided to attach O$_1$ only to the state that corresponds to the completion of the first gating step with no further transitions (neither the second voltage-sensor transition nor the gating ring transition) in all four voltage sensors. In contrast, to enter O$_2$, all subunits must complete both voltage sensor transitions and the collective gating ring transition. This model's structure is displayed in Supp. 1 to *Figure 9*.

We also tested models in which the upper floor was only accessible from a subset of states and models with O$_1$ attached to more than a single closed state or even multiple O$_1$ states with varying conductance. While those more complex models offered a gradual improvement matching experimental traces, they showed no striking advantages over the more symmetric and parsimonious model presented here. Furthermore, we have extensively tested variants with a broken symmetry in the gating kinetics. In these models, the conventional gating steps (rates α, β, γ, δ) differed between the ground floor and the newly introduced top floor states, e.g., by the introduction of factors to the α, β, γ, δ values. Under these conditions, a local balance was achieved by corresponding inverse factors to the local κ and $\lambda$. Again, the gradual improvements that could be achieved did not seem to justify the introduction of the additional parameters.

## Voltage dependence of rates

The simplest model structure that allowed us to reproduce the key observations (*Figure 9—figure supplement 1*) was implemented in Igor Pro as a system of coupled differential equations that govern the mean-field occupancies of the 33 states. To simulate the stochastic gating of a single channel, the Gillespie Algorithm was implemented (*Gillespie, 1976*). For the voltage-dependent transition rates, we chose a sigmoidal functional form (*Equation 5*).

$$x = x_0 + \frac{A_x}{1 + \exp\left(\left(V_m - V_x^h\right) \cdot k_x\right)} \tag{5}$$

where x stands for one of the rates α, β, γ, δ, that govern transitions in the first and second dimension of the model, which we attribute to two sequential steps of voltage sensor displacement. The rates $\eta$ and θ of opening and closing of the mutant-specific open state O$_1$ also follow this functional form, as does the rate $\lambda$, governing the return from the upper layer. The rate κ of entering the upper layer is the sum of two components, $\kappa_{all} = \kappa_l + \kappa_r$, which again follow the same functional form. The rates leading towards and away from the second open state O$_2$, ε, $\zeta$, r$_{open}$ and r$_{close}$ are constants.

## Acknowledgements

We thank the expert technical assistance of Kerstin Dümke. RA was supported by an IMPRS Neurosciences scholarship. This work was supported by Deutsche Forschungsgemeinschaft (DFG, German Research Foundation) Project-ID 436260547, in relation to NeuroNex (NSF 2015276) (to AN)

## Additional information

### Funding

| Funder | Grant reference number | Author |
|---|---|---|
| Max-Planck-Gesellschaft | | Reham Abdelaziz<br>Adam P Tomczak<br>Luis A Pardo |
| International Max Planck Research School for Neurosciences | | Reham Abdelaziz |
| Deutsche Forschungsgemeinschaft | 436260547 | Andreas Neef |

The funders had no role in study design, data collection and interpretation, or the decision to submit the work for publication.The funders had no role in study design, data collection and interpretation, or the decision to submit the work for publication.

### Author contributions

Reham Abdelaziz, Conceptualization, Formal analysis, Investigation, Writing – original draft, Writing – review and editing; Adam P Tomczak, Investigation, Writing – review and editing; Andreas Neef, Conceptualization, Software, Formal analysis, Writing – original draft, Writing – review and editing; Luis A Pardo, Conceptualization, Formal analysis, Funding acquisition, Investigation, Writing – original draft, Project administration, Writing – review and editing

### Author ORCIDs

Andreas Neef (iD) https://orcid.org/0000-0003-4445-7478
Luis A Pardo (iD) https://orcid.org/0000-0003-1375-4349

### Ethics

Xenopus laevis oocytes were obtained following the protocol approved by the Lower Saxony State Office for Consumer Protection and Food Safety (permit 33.9-42502-05-20A520). Surgery was performed under tricaine anesthesia and both local and systemic analgesia were provided during and after surgery. Every effort was made to minimize suffering of the animals.

Reviewer #1 https://doi.org/10.7554/eLife.91420.4.sa1
Author response https://doi.org/10.7554/eLife.91420.4.sa2

## Additional files

### Supplementary files

• Supplementary file 1. Parameters of a global fit that linked the first component of the biphasic response.

• Supplementary file 2. Model parameters.

• Supplementary file 3. Primers used for infusion cloning or site-directed mutagenesis. The sequences are listed 5'–3'. For mutagenesis primers, only the sense sequences are given. The reverse primers corresponded to the reverse-complement sequence.

• MDAR checklist

## Data availability

All data generated or analysed during this study are included in the manuscript and supporting files. Source data is provided for Figure 1B,D, Figure 2A,D, Figure 3B, Figure 6B, C, Figure 7 A-D, and Figure 8C, D. Modeling code and complete reproduction of all model figures is available at https://doi.org/10.17617/3.GQAPEU.

The following dataset was generated:

| Author(s) | Year | Dataset title | Dataset URL | Database and Identifier |
|---|---|---|---|---|
| Andreas N | 2024 | Revealing a hidden conducting state in KV10.1 mutants | https://doi.org/10.17617/3.GQAPEU | Edmond, 10.17617/3.GQAPEU |

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

# Appendix 1

## Appendix 1—key resources table

| Reagent type (species) or resource | Designation | Source or reference | Identifiers | Additional information |
|---|---|---|---|---|
| Gene (*Homo sapiens*) | Kv10.1 | NA | NCBI NM_002238.4 | |
| Antibody | anti-Myc | SIGMA | M4439, RRID:AB_439694 | 3 µg IP, 1:1000 immunoblot |
| Antibody | anti-Myc | Abcam | ab206486; RRID:AB_2861226 | 3 µg IP |
| Antibody | anti-Kv10.1 | *Chen et al., 2011* | polyclonal anti-Kv10.1 | 1:1500 |
| Recombinant DNA reagent | pSGEM Kv10.1 | Addgene #85704 | | |
| Chemical compound, drug | Thapsigargin | Abcam | ab120286 | 5 µM |
| Chemical compound, drug | Ionomycin | Abcam | ab120116 | 5 µM |
| Chemical compound, drug | Astemizole | Esteve Química | N/A | 100 µM |
| Software, algorithm | Patch Master | HEKA | | |
| Software, algorithm | FitMaster | HEKA | | |
| Software, algorithm | Igor Pro | WaveMetrics | | |

