## [Editor Report · eLife assessment]

This **valuable** study examines the role of the interaction between cytoplasmic N- and C-terminal domains in voltage-dependent gating of Kv10.1 channels. The authors suggest that they have identified a hidden open state in Kv10.1 mutant channels, thus providing a window for observing early conformational transitions associated with channel gating. The evidence supporting the major conclusions is **solid**, but additional work is required to determine the molecular mechanism underlying the observations in this study. Learning the molecular mechanisms could be significant in understanding the gating mechanisms of the KCNH family and will appeal to biophysicists interested in ion channels and physiologists interested in cancer biology.

---

## [Referee Report · Reviewer #1]

Gating of Kv10 channels is unique because it involves coupling between non-domain swapped voltage sensing domains, a domain-swapped cytoplasmic ring assembly formed by the N- and C-termini, and the pore domain. Recent structural data suggests that activation of the voltage sensing domain relieves a steric hindrance to pore opening, but the contribution of the cytoplasmic domain to gating is still not well understood. This aspect is of particular importance because proteins like calmodulin interact with the cytoplasmic domain to regulate channel activity. The effects of calmodulin (CaM) in WT and mutant channels with disrupted cytoplasmic gating ring assemblies are contradictory, resulting in inhibition or activation, respectively. The underlying mechanism for these discrepancies is not understood. In the present manuscript, Reham Abdelaziz and collaborators use electrophysiology, biochemistry and mathematical modeling to describe how mutations and deletions that disrupt inter-subunit interactions at the cytoplasmic gating ring assembly affect Kv10.1 channel gating and modulation by CaM. In the revised manuscript, additional information is provided to allow readers to identify within the Kv10.1 channel structure the location of E600R, one of the key channel mutants analyzed in this study. However, the mechanistic role of the cytoplasmic domains that this study focuses on, as well as the location of the ΔPASCap deletion and other perturbations investigated in the study remain difficult to visualize without additional graphical information.

The authors focused mainly on two structural perturbations that disrupt interactions within the cytoplasmic domain, the E600R mutant and the ΔPASCap deletion. By expressing mutants in oocytes and recording currents using Two Electrode Voltage-Clamp (TEV), it is found that both ΔPASCap and E600R mutants have biphasic conductance-voltage (G-V) relations and exhibit activation and deactivation kinetics with multiple voltage-dependent components. Importantly, the mutant-specific component in the G-V relations is observed at negative voltages where WT channels remain closed. The authors argue that the biphasic behavior in the G-V relations is unlikely to result from two different populations of channels in the oocytes, because they found that the relative amplitude between the two components in the G-V relations was highly reproducible across individual oocytes that otherwise tend to show high variability in expression levels. Instead, the G-V relations for all mutant channels could be well described by an equation that considers two open states O1 and O2, and a transition between them; O1 appeared to be unaffected by any of the structural manipulations tested (i.e. E600R, ΔPASCap, and other deletions) whereas the parameters for O2 and the transition between the two open states were different between constructs. The O1 state is not observed in WT channels and is hypothesized to be associated with voltage sensor activation. O2 represents the open state that is normally observed in WT channels and is speculated to be associated with conformational changes within the cytoplasmic gating ring that follow voltage sensor activation, which could explain why the mutations and deletions disrupting cytoplasmic interactions affect primarily O2.

Severing the covalent link between the voltage sensor and pore reduced O1 occupancy in one of the deletion constructs. Although this observation is consistent with the hypothesis that voltage-sensor activation drives entry into O1, this result is not conclusive. Structural as well as functional data has established that the coupling of the voltage sensor and pore does not entirely rely on the S4-S5 covalent linker between the sensor and the pore, and thus the severed construct could still retain coupling through other mechanisms, which is consistent with the prominent voltage dependence that is observed. If both states O1 and O2 require voltage sensor activation, it is unclear why the severed construct would affect state O1 primarily, as suggested in the manuscript, as opposed to decreasing occupancy of both open states. In line with this argument, the presence of Mg2+ in the extracellular solution affected both O1 and O2. This finding suggests that entry into both O1 and O2 requires voltage-sensor activation because Mg2+ ions are known to stabilize the voltage sensor in its most deactivated conformations.

Activation towards and closure from O1 is slow, whereas channels close rapidly from O2. A rapid alternating pulse protocol was used to take advantage of the difference in activation and deactivation kinetics between the two open components in the mutants and thus drive an increasing number of channels towards state O1. Currents activated by the alternating protocol reached larger amplitudes than those elicited by a long depolarization to the same voltage. This finding is interpreted as an indication that O1 has a larger macroscopic conductance than O2. In the revised manuscript, the authors performed single-channel recordings to determine why O1 and O2 have different macroscopic conductance. The results show that at voltages where the state O1 predominates, channels exhibited longer open times and overall higher open probability, whereas at more depolarized voltages where occupancy of O2 increases, channels exhibited more flickery gating behavior and decreased open probability. These results are informative but not conclusive since single-channel amplitudes could not be resolved at strong depolarizations, limiting the extent to which the data could be analyzed. In the last revision, the authors have included one representative example showing inhibition of single channel activity by the Kv10-specific inhibitor astemizole. Group data analysis would be needed to conclusively establish that the currents that were recorded indeed correspond to Kv10 channels.

It is shown that conditioning pulses to very negative voltages result in mutant channel currents that are larger and activate more slowly than those elicited at the same voltage but starting from less negative conditioning pulses. In voltage-activated curves, O1 occupancy is shown to be favored by increasingly negative conditioning voltages. This is interpreted as indicating that O1 is primarily accessed from deeply closed states in which voltage sensors are in their most deactivated position. Consistently, a mutation that destabilizes these deactivated states is shown to largely suppress the first component in voltage-activation curves for both ΔPASCap and E600R channels.

The authors then address the role of the hidden O1 state in channel regulation by calcium-calmodulin (CaM). Stimulating calcium entry into oocytes with ionomycin and thapsigargin, assumed to enhance CaM-dependent modulation, resulted in preferential potentiation of the first component in ΔPASCap and E600R channels. This potentiation was attenuated by including an additional mutation that disfavors deeply closed states. Together, these results are interpreted as an indication that calcium-CaM preferentially stabilizes deeply closed states from which O1 can be readily accessed in mutant channels, thus favoring current activation. In WT channels lacking a conducting O1 state, CaM stabilizes deeply closed states and is therefore inhibitory. It is found that the potentiation of ΔPASCap and E600R by CaM is more strongly attenuated by mutations in the channel that are assumed to disrupt interaction with the C-terminal lobe of CaM than mutations assumed to affect interaction with the N-terminal lobe. These results are intriguing but difficult to interpret in mechanistic terms. The strong effect that calcium-CaM had on the occupancy of the O1 state in the mutants raises the possibility that O1 can be only observed in channels that are constitutively associated with CaM. To address this, a biochemical pull-down assay was carried out to establish that only a small fraction of channels are associated with CaM under baseline conditions. These CaM experiments are potentially very interesting and could have wide physiological relevance. However, the approach utilized to activate CaM is indirect and could result in additional non-specific effects on the oocytes that could affect the results.

Finally, a mathematical model is proposed consisting of two layers involving two activation steps for the voltage sensor, and one conformational change in the cytoplasmic gating ring - completion of both sets of conformational changes is required to access state O2, but accessing state O1 only requires completion of the first voltage-sensor activation step in the four subunits. The model qualitatively reproduces most major findings on the mutants. Although the model used is highly symmetric and appears simple, the mathematical form used for the rate constants in the model adds a layer of complexity to the model that makes mechanistic interpretations difficult. In addition, many transitions that from a mechanistic standpoint should not depend on voltage were assigned a voltage dependence in the model. These limitations diminish the mechanistic insight that can be reliably extracted from the model.

---

## [Author Response]

The following is the authors’ response to the previous reviews.

We appreciate the feedback provided and refer to our previous response for detailed explanations regarding our decisions on some of the recommendations made by the referees and editors. We have introduced changes as follows:

• We added a supplementary Figure to Figure 5 to show inhibition by Astemizole at the single channel level.

• We have corrected Figure 7A, where the normalized current did not reach 1 as a maximum. We had overlooked that this is expected when the prepulse was -160 mV, and the IV is strongly biphasic, but not when coming from -100 mV. We are thankful for this observation, which served to identify that the values for one of the cells were inverted with respect to the others (the sequence of stimuli was different during recording, and this information got lost in the analysis procedure). We have corrected this and made sure that such a mistake had not happened anywhere else.

• Finally, we have corrected a typo in the discussion, as indicated in the review.

We include a version with changes marked and a clean version of the manuscript.